# Research

theoretical biology, ecology, immunology

cellular immunotherapy, predator–prey dynamics, CAR T cells, stochastic process

**Authors for correspondence:**
Frederick L. Locke
e-mail: frederick.locke@moffitt.org
Philipp M. Altrock
e-mail: philipp.altrock@gmail.com

†These authors contributed equally to the study.

# The roles of T cell competition and stochastic extinction events in chimeric antigen receptor T cell therapy

Gregory J. Kimmel[1], Frederick L. Locke[2,†] and Philipp M. Altrock[1,2,3,†]

[1]Department of Integrated Mathematical Oncology, [2]Department of Blood and Marrow Transplant and Cellular Immunotherapy, and [3]Department of Malignant Hematology, H. Lee Moffitt Cancer Center and Research Institute, Tampa, FL, USA

GJK, 0000-0001-9766-5399; PMA, 0000-0001-7731-3345

Chimeric antigen receptor (CAR) T cell therapy is a remarkably effective immunotherapy that relies on *in vivo* expansion of engineered CAR T cells, after lymphodepletion (LD) by chemotherapy. The quantitative laws underlying this expansion and subsequent tumour eradication remain unknown. We develop a mathematical model of T cell–tumour cell interactions and demonstrate that expansion can be explained by immune reconstitution dynamics after LD and competition among T cells. CAR T cells rapidly grow and engage tumour cells but experience an emerging growth rate disadvantage compared to normal T cells. Since tumour eradication is deterministically unstable in our model, we define cure as a stochastic event, which, even when likely, can occur at variable times. However, we show that variability in timing is largely determined by patient variability. While cure events impacted by these fluctuations occur early and are narrowly distributed, progression events occur late and are more widely distributed in time. We parameterized our model using population-level CAR T cell and tumour data over time and compare our predictions with progression-free survival rates. We find that therapy could be improved by optimizing the tumour-killing rate and the CAR T cells' ability to adapt, as quantified by their carrying capacity. Our tumour extinction model can be leveraged to examine why therapy works in some patients but not others, and to better understand the interplay of deterministic and stochastic effects on outcomes. For example, our model implies that LD before a second CAR T injection is necessary.

## 1. Introduction

In 2017, non-Hodgkin lymphoma was the most common haematologic malignancy in the US with 72 000 new cases (4.3% of all cancer) and 20 000 deaths (3.4% of all cancer deaths) [1]. Large B cell lymphoma (LBCL) is the most common subtype of non-Hodgkin lymphoma. LBCL arises in the B cell lineage for which the transmembrane protein CD19 is a specific marker. Historically, LBCL patients that did not respond to chemotherapy have a median overall survival of under seven months [2]. These patients could benefit from autologous chimeric antigen receptor (CAR) T cell therapy that uses genetically engineered T cells specifically re-targeted to CD19 [3]. A pivotal, multi-centre, phase 1–2 trial of the CAR T cell drug axicabtagene ciloleucel (axi-cel; $n = 101$ patients treated) was ZUMA-1 [4,5]. Overall response rate and complete response rate in ZUMA-1 were 82% and 54%. The respective responses to standard chemotherapy are 26% and 7% [2]. While many LBCL patients treated with this cellular therapy have seen a temporary reduction in tumour burden, about 60% eventually progress. A complete understanding of why these patients progress is lacking.

Cellular immunotherapies, such as CAR T cell therapy, encompass a new frontier for predictive mathematical biological modelling [6–9]. One of the

first goals of this new field is to describe and predict CAR T cell expansion and decay after administration. Recent works used an empirical time-dependent modelling approach and compartment modelling [6] to describe the complicated temporal kinetics of the CAR T cell drug tisagenlecleucel [10]. Others sought to quantify ecological dynamics of CAR T cells to explain expansion and exhaustion [8], and signalling-induced cell state variability [9], both using *in vitro* data. Current modelling has not considered interactions between CAR and normal T cells, nor paid much attention to feedback between tumour and CAR T cells [11–13].

Here, we seek to better understand T and CAR T cell dynamics, and the resulting tumour cell dynamics *in vivo* using mathematical modelling. We bin the potentially different CAR T cell phenotypes together [14,15], and model selection in the T cell homeostatic niche by including normal T cells. Tumour dynamics can be stochastic due to low cell counts. Our framework explains the overall CAR kinetics and reveals that stochastic dynamics in small tumours match clinical progression.

## 2. Methods

We model dynamics and interactions among normal T cells, CAR T cells and tumour cells. The model considers three cell populations in the form of continuous-time birth and death stochastic processes and their deterministic mean-field equations: normal T cells, $N$, CAR T cells, $C$, both given in cells $\mu l^{-1}$, and antigen-presenting tumour cells, $B$, given in ml. These measurements can be converted to cell counts. A more detailed overview of the methods can be found in the electronic supplementary material, sections 1 and 2.

We define the following two scenarios. Complete response (CR) is achieved when the tumour is eradicated. Progressive disease (PD) is counted when the tumour has reached 120% of its initial size. Note that the clinical definition of progression typically is less quantitatively tractable [16], as clinical progression can occur whenever the disease worsens in relation to its nadir size, which could be markedly less than the value at baseline [17].

We consider the case of lymphodepleting chemotherapy prior to infusion of autologous CAR T cells. We set time to 0 at the time of CAR infusion. Normal and CAR T cell populations then grow towards their respective carrying capacities but influence each other. This mutual influence gives rise to selection.

We exclude possible influences by other sources of CD19, such as normal B cells, based on the following observations. Patients have effectively zero normal B cells following lymphodepleting chemotherapy and prior/concurrent to CAR T cell infusion. Furthermore, the reconstitution of normal B cells is extremely slow, such that only 50% of patients have recovered any normal B cells by 1 year after therapy. While it is possible that normal B cells activate CD19 directed CAR T cells, at this scale the dynamics of CAR T are probably not impacted by normal B cells. Thus, in the patient population we are interested in, normal sources of CD19 do not seem to play a significant role. Meanwhile, the tumour cell population $B$ grows autonomously at a net growth rate $r_B$ and experiences tumour killing at rate $\gamma_B$, proportional to the number of CAR T cells.

These biological mechanisms could manifest themselves in multiple ways in a mathematical model. A carrying capacity for T cells could emerge due to predation, resource or spatial limitations, birth and death rate balance through exogeneous effects (e.g. paracrine signalling [18]), or any combination of the above. We investigated different functional forms (a generalized logistic, Gompertz, and an explicit interaction model,

discussed in the electronic supplementary material, section 3). Using an information criterion, we elected to use a Gompertz model approach. The corresponding dynamical system in the mean-field limit is

$$\frac{\mathrm{d}N}{\mathrm{d}t} = -r_N\, N \ln\left[\frac{N+C}{K_N}\right], \tag{2.1}$$

$$\frac{\mathrm{d}C}{\mathrm{d}t} = -r_C(T)\, C \ln\left[\frac{N+C}{K_C}\right] \tag{2.2}$$

and $$\frac{\mathrm{d}B}{\mathrm{d}t} = r_B B - \gamma_B\, B \frac{C}{k_B + C}. \tag{2.3}$$

Here $T = N + C$ is the total lymphocyte count, and $r_C(T) = \rho_C + b(T - K_N)^2/(a\, T^2 + (T - K_N)^2)$, where $\rho_C$ is a background expansion rate and the second term reflects that growth can be muted when the overall (largely normal) T cell population reaches capacity, modulated by the two parameters $a$ and $b$. We introduced feedback of total lymphocyte count on CAR expansion because the CAR T cell population expands at a faster rate initially and contracts slower after peak, which has been a challenge in several of the recent quantitative modelling approaches [6,11]. A generalized logistic or Gompertz form cannot alone capture this behaviour. Thus, we chose to include additional feedback into the intrinsic growth rate function $\rho_C$, which becomes a function of the total T cell count $T$, the tumour mass $B$, or both (see electronic supplementary material, section 3). For a deeper discussion on the assumptions and details of the system of equations (2.1)–(2.3), which can be based on a stochastic birth and death process, see the electronic supplementary material, section 3, where we also present a linear stability analysis. Figure 1a shows a schematic of the dynamical system. Figure 1b shows the associated cellular events, which can be interpreted deterministically or stochastically. The deterministic system's *qualitative* behaviour aligns with clinical observations (figure 1d–f).

Tumour killing by CAR T cells is modelled as a result of contact, thus is proportional to $B \times C$ and includes a saturation factor, motivated by the assumption that there exists an upper limit at which CAR T cells can interact and kill the tumour cells.

Altogether, these assumptions lead to seven parameters to be fitted by optimization, based on 11 data points from clinical observations—five for CAR T and five for absolute lymphocyte count (ALC), excluding the initial conditions, and one measurement for the tumour state at day 30 (for further details see electronic supplementary material, section 4).

## 3. Results

We calibrated the deterministic model using clinical data (figure 2a,b; electronic supplementary material, sections 2 and 4), which led to the parameters in table 1. To recapitulate and predict progression-free survival over time, we employed the corresponding stochastic formulation (electronic supplementary material, section 5), which is able to capture dynamics of small tumour populations near extinction.

### (a) Parameter sensitivity and identifiability

We conducted a local sensitivity and identifiability analysis [21–23] of the fitted parameters. The resulting ranking of parameters highlights that the carrying capacities of the CAR and normal T cells, respectively, and the maximal tumour-killing rate were most sensitive (figure 2c). Note here that low sensitivity of a parameter does not imply that the model is not sensitive to these parameters. We performed an identifiability analysis, with a threshold value of 0.9, to indicate non-identifiability between parameters pairs. Strong Pearson

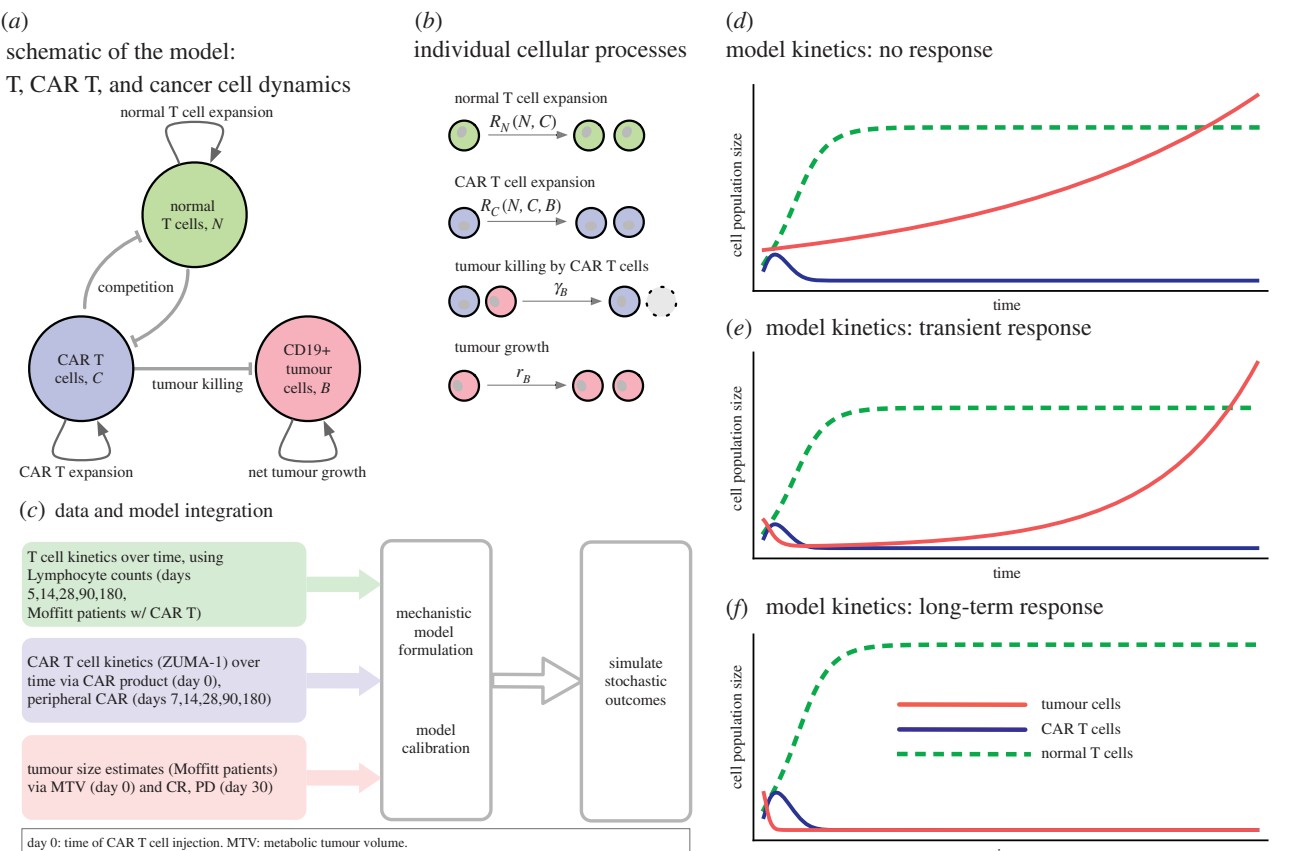

**Figure 1.** Overview of cellular interactions, kinetics, data integration and qualitative dynamics. (*a*) Model schematic of three cell compartments: CAR T cells, *C*, proliferate and interact with resident (normal) lymphocytes, *N* (depleted by lymphodepleting chemotherapy). CAR T cells also engage in killing CD19+ tumour and other *B* cells. (*b*) The system can be described by four cellular kinetic reactions, with density-dependent feedback (differences in carrying capacities) in the net expansion rates: $R_X = r_X \ln(K_X/(N+C))$, where *X* stands for *N* or *C*. The deterministic, large population size limit of these dynamics is given by equations (2.1)–(2.3). (*c*) Schematic of data integration to parametrize the mathematical model; we used longitudinal data of peripheral absolute lymphocyte count (ALC), peripheral CAR+ cell counts per µL, and the tumour volume-changes as estimated from patients of the ZUMA-1 trial with complete response (CR) or progressive disease (PD). We assumed that, at days 30, 60 or 90, CRs had no detectable tumour mass, and that PDs had 1.2 times their initial tumour size. Median initial tumour mass was 94.86 ml. The qualitative dynamics of the system, given by equations (2.1)–(2.3): no response to CAR therapy (*d*), transient response followed by progression/relapse (*e*) and long-term response (tumour appears to be eradicated) (*f*). These illustrative examples are not to scale.

correlations between $k_B$ and $r_B$ (0.94), and between $k_B$ and $\gamma_B$ (−0.96) were observed (figure 2*d* inset), which suggests that these pairs are not identifiable. All other pairs were below the threshold and can be considered identifiable. The sensitivity ranking suggests that the most promising avenues to improve CAR T cell therapy should focus on improving adaptability (carrying capacity) and efficacy (tumour-killing rate) of the engineered CAR T cell product.

## (b) Immune reconstitution and return to homeostasis during chimeric antigen receptor T cell therapy follows a Gompertz growth law

We initially hypothesized that immune reconstitution follows a generalized logistic growth equation of the form $x'(t) = rx(1 − (x/k)^c)$ [24]. This approach contains the standard logistic model ($c = 1$) and leads to Gompertz growth in the limits $c \to 0$, $r \to \infty$. Our nonlinear optimization routine for data fitting (see electronic supplementary material, section 3) selected Gompertz as the best approach for immune reconstitution dynamics (figure 2*a*). The optimizer was selecting *r* at the edge of the upper boundary of its optimization region, while selecting *c* at the lowest possible (positive) value. We

concluded that immune reconstitution follows Gompertz growth, $x'(t) = −rx\ln[x/k]$.

## (c) Chimeric antigen receptor T cell kinetics can be explained by increased growth rate and lower carrying capacity

We hypothesized that CAR T cells become maladapted during the manufacturing process, lowering their carrying capacity. After lymphodepletion, a complex signalling cascade occurs that results in immune reconstitution towards homeostatic levels [19], driven by stem and progenitor cells [25,26]. Both normal T cells (figure 2*a*) and CAR T cells (figure 2*b*) use the changing environment to proliferate and expand, but only normal cells can be reconstituted from stem cells. Although we do not model these signals explicitly, the system's behaviour can be observed through the interplay of the normal and CAR T cells. As a result, CAR T cells are outcompeted by normal T cells [27] in the long term, resulting in lower carrying capacity. An alternative hypothesis that could explain the kinetics would be predation of CAR T cells by normal T cells. We examined and ultimately discarded this alternative hypothesis

Proc. R. Soc. B 288: 20210229

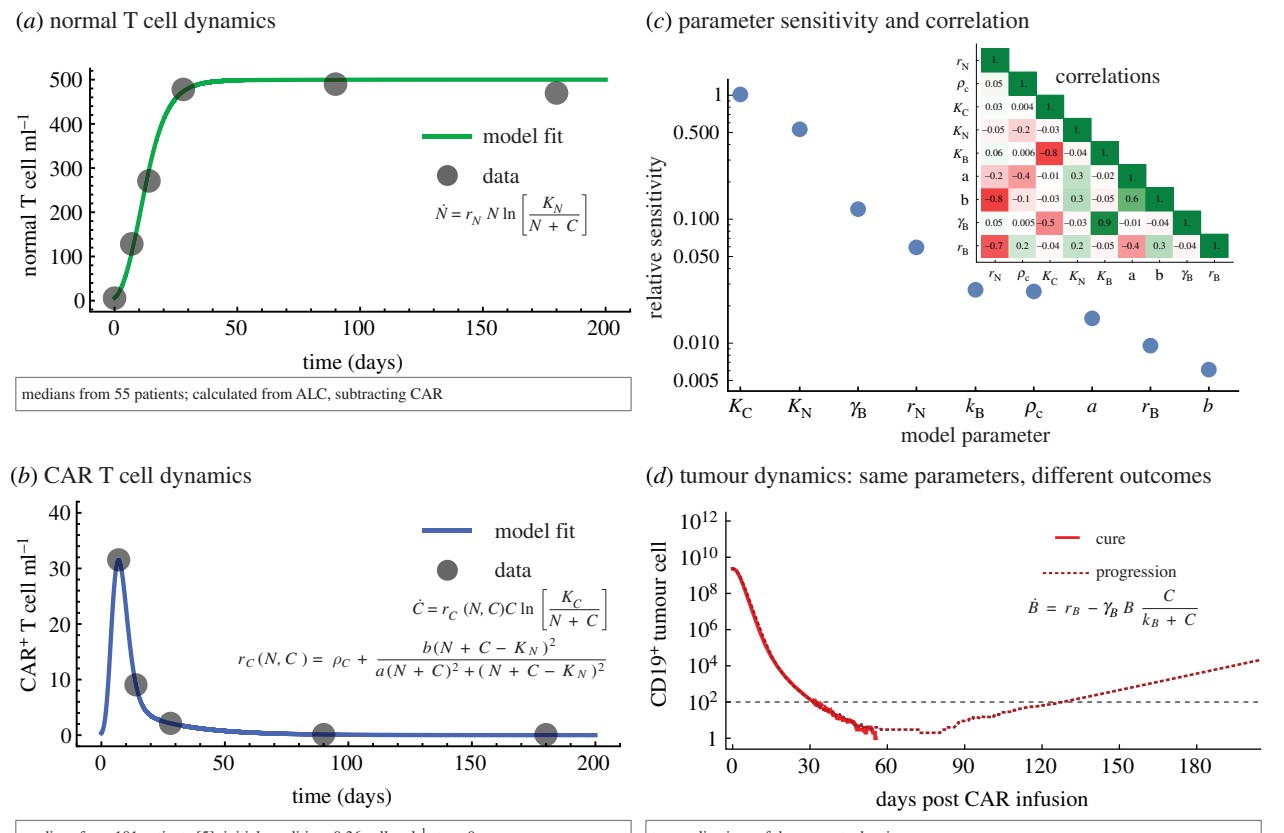

**Figure 2.** Comparison of model fits and data in the T cell compartments, stochastic dynamics in the tumour cell compartment. (*a*) ALC was used to parameterize normal T cell dynamics, equation (2.1), using ALC-CAR from peripheral blood to estimate normal T cell counts $N$. (*b*) CAR positive T cell dynamics, equation (2.2), were parameterized using ZUMA-1 trial data of median peripheral CAR counts, to fit peak and decay of CAR. Nonlinear optimization for data fitting explained in the electronic supplementary material, section 4. (*c*) Parameters ranked by local relative sensitivity measure (electronic supplementary material, section 4). Inset: parameter correlation matrix (Pearson correlation, see electronic supplementary material). (*d*) Two example trajectories of tumour burden over time, using identical parameters and initial conditions for the hybrid deterministic-stochastic process in the tumour compartment. Both examples enter the stochastic region (less than 100 tumour cells), but one escapes leading to progression. All parameter values and initial conditions used are given in table 1. Hybrid model simulation procedure described in the electronic supplementary material, section 5. (Online version in colour.)

**Table 1.** Parameter and initial condition values as identified by our machine learning procedure and from the literature (also see electronic supplementary material, section 4). The CAR T cell growth rate function is $r_C$, which depends on parameters $\rho_C$, a, b.

| biological parameter | symbol | fitted value | quartile range | ref. |
|---|---|---|---|---|
| normal T cell carrying capacity | $K_N$ | $2.50 \times 10^{11}$ cells | n/a | [19] |
| CAR T cell carrying capacity | $K_C$ | $6.96 \times 10^{10}$ cells | $[6.15, 9.65] \times 10^{10}$ cells | this work |
| normal T cell net growth rate | $r_N$ | $1.70 \times 10^{-1}$ day$^{-1}$ | $[1.65, 1.70] \times 10^{-1}$ day$^{-1}$ | this work |
| baseline CAR T net growth rate | $\rho_C$ | $2.51 \times 10^{-2}$ day$^{-1}$ | $[2.08, 3.54] \times 10^{-2}$ day$^{-1}$ | this work |
| signalling inefficiency factor in $r_C$ | $a$ | $4.23 \times 10^{-1}$ | $[1.00, 3.02] \times 10^{-1}$ | this work |
| immune reconstitution impact in $r_C$ | $b$ | $5.25 \times 10^{-1}$ day$^{-1}$ | $[4.67, 5.22] \times 10^{-1}$ day$^{-1}$ | this work |
| tumour net growth rate | $r_B$ | $(1 - 50) \times 10^{-2}$ day$^{-1}$ | n/a | [20] |
| tumour-killing rate (by effector CAR) | $\gamma_B$ | $1.15 \times 10^{0}$ day$^{-1}$ | $[0.64, 1.35] \times 10^{0}$ day$^{-1}$ | this work |
| killing rate saturation parameter | $k_B$ | $2.024 \times 10^{9}$ cells | $[1.40, 3.125] \times 10^{9}$ cells | this work |
| initial median normal T cell number | $N(t=0)$ | $3.00 \times 10^{9}$ cells | n/a | [5] |
| initial CAR T cell number | $C(0)$ | $1.80 \times 10^{8}$ cells | n/a | [5] |
| initial median tumour cell number | $B(0)$ | $9.486 \times 10^{10}$ cells | n/a | [5] |

after statistical examination (see electronic supplementary material, section 3).

CAR T cell persistence, at least for some time, could result from CAR T effector memory cells [6]. Currently, it is unclear

whether CAR T persistence plays a role in LBCL treatment [4]; remarkably, cure is possible without CAR T cell persistence. The decline over time of CAR T cells may occur on a time frame longer than the expected survival time, even in patients

with tumour extinction. Therefore, the clinical definition of (temporal) CAR T persistence may be captured by our model.

CAR T expansion occurs rapidly within the first two weeks. By contrast, CAR T cells decay on a much slower time scale. For instance, the median patient still retains levels comparable to their value at infusion by day 90. Hence, other functional dependencies need to be placed on the CAR T cell growth rate (equation (2.2)). The baseline rate $\rho_C$ sets the time scale at which birth and death events occur on average in the CAR T cell population. These birth and death rates depend on signals provided by all T cells, which change during therapy as the immune system returns to homeostatic levels. Hence, we expect the overall CAR T cell turnover rate to change during the course of reconstitution. The dependence on the total lymphocyte count $T$ describes these nonlinear dynamics as the total T cell count approaches a carrying capacity, $T \to K_N$.

## (d) Stochastic tumour extinction

As a consequence of CAR T cell impairment in renewal capacity, tumour eradication is deterministically unstable and not a long-term outcome (see electronic supplementary material, section 3). However, tumour mass often shrinks at least for some time during treatment and can temporarily be brought down to very low levels [4], leading to possible stochastic extinction (figure 2d). We predict that if cure occurs, it does so via a stochastic event in which the malignant B cells are driven to extinction [28]. This stochastic approach leads to the question of whether parameter variability (e.g. patient variability), or stochasticity of the underlying process, or both, are responsible for the broad distributions of the times to cure or progression and probability of cure. Of those ZUMA-1 patients that were treated at Moffitt, we assessed the variability in CAR T cell kinetics, although not all patients had all time points available (figure 3a). The Moffitt ZUMA-1 patient cohort's (overall $n = 23$ patients) variabilty in CAR T count (cells $\mu l^{-1}$) were 27.5 (IQR: 16.5,57.5; $n = 19$), 9.1 (IQR: 3.8,31.1; $n = 23$), 2.1 (IQR: 0.6,6.1; $n = 22$) and 0.1 (IQR: 0.01,0.4; $n = 22$) at days 7, 14, 28 and 90, respectively.

## (e) Probability and time to progression

To evaluate our model parameterization, we compared the overall progression-free survival (PFS) curve for all patients in the ZUMA-1 [5] trial to a virtual cohort of 1000 simulated patients with varying tumour growth rates (figure 3b). Although we had not used progression-free survival as a goal function to find suitable model parameters (Methods, electronic supplementary material, section 4), our stochastic model recapitulates PFS of the ZUMA-1 trial for reasonable tumour growth rate values, using a hybrid deterministic–stochastic numerical approach (electronic supplementary material, section 5) in which the system is simulated deterministically if the tumour is above a threshold value, and the tumour is simulated stochastically once below this threshold (the outcomes were very weakly impacted by choice of threshold; see electronic supplementary material). As a result, a probability of tumour extinction can be calculated numerically as a function of specific model parameters for a fixed point in time or overall. Treatment success (probability of tumour extinction) critically depends on the ability of the CAR T to survive (figure 3c), and on the effectiveness of lymphodepletion that reduce absolute lymphocyte counts (figure 3d).

## (f) Variability in outcomes

A natural question involves whether overall variability in timing to cure or progression is shaped by patient variability (implemented using a hyperparameter that perturbs the parameters, see electronic supplementary material, section 5.3) in contrast with model stochastic effects. We performed additional stochastic simulations of tumour extinction (electronic supplementary material, section 5.4) under controlled parameter variation. We found that the probability of cure changes with overall parameter variability (figure 3e). Similarly, the time to cure distribution for a specific set of parameters widens with increasing parameter variability (figure 3f). Thus, variation in patient-specific conditions and parameter values could be the main determinant of observed variability in timing of cure or progression.

Most stochastic simulations resulted in cure between days 20 and 80. We rarely found late cure events up to day 140 (figure 4a). Meanwhile, progression times were distributed over a broader range. Typical progression times, as defined in our model as 120% of initial tumour burden, occur anywhere between days 200 and 500 (figure 4b). These large differences in time scales occur because cure, as a stochastic tumour extinction event, is much more likely to occur before CAR T cells begin to decline, typically after day 14.

## (g) Necessity of lymphodepletion

In the context of timing of events, our model can be useful to test new treatment strategies *in silico*, to inform clinical trial design. For example, we used our model to inform the timing of a second infusion, with or without additional lymphodepletion (electronic supplementary material, section 6) [11]. To improve outcomes with a second infusion, lymphodepletion that resets the T cells is necessary. This suggests that a second, lower dose lymphodepletion alone might be sufficient, provided it does not kill all CAR T cell but lowers overall T cell density sufficiently. The temporal suppression of normal lymphocytes is a key driver of CAR expansion, which, together with transient tumour burden data, could be leveraged to further evaluate the benefit of second interventions.

## 4. Discussion

Here, we propose a modelling framework for the analysis and prediction of cellular kinetics during CAR T cell therapy. We focus on normal T cells, CAR T cells and tumour cells and find that CAR expansion and decay can be explained via competitive growth in the context of immune reconstitution, which is a consequence of the lymphodepletion prior to therapy. A ramification of the model is that cure must be a stochastic event. However, the likelihood of cure is largely determined by specific parameters and tumour fluctuations play a minor role in the variability of clinical outcomes. These insights can be leveraged to better understand why therapy works for some but not all, and how it can be improved.

We posited four potential drivers of patient outcomes: the effects of normal T cell dynamics on CAR T cells, CAR T cell expansion (peak), CAR T cell durability (slow decay) and tumour-killing rate. To better understand these processes, we developed a cell population-ecological framework that describes the kinetics of normal T, CAR T and tumour cells. We performed

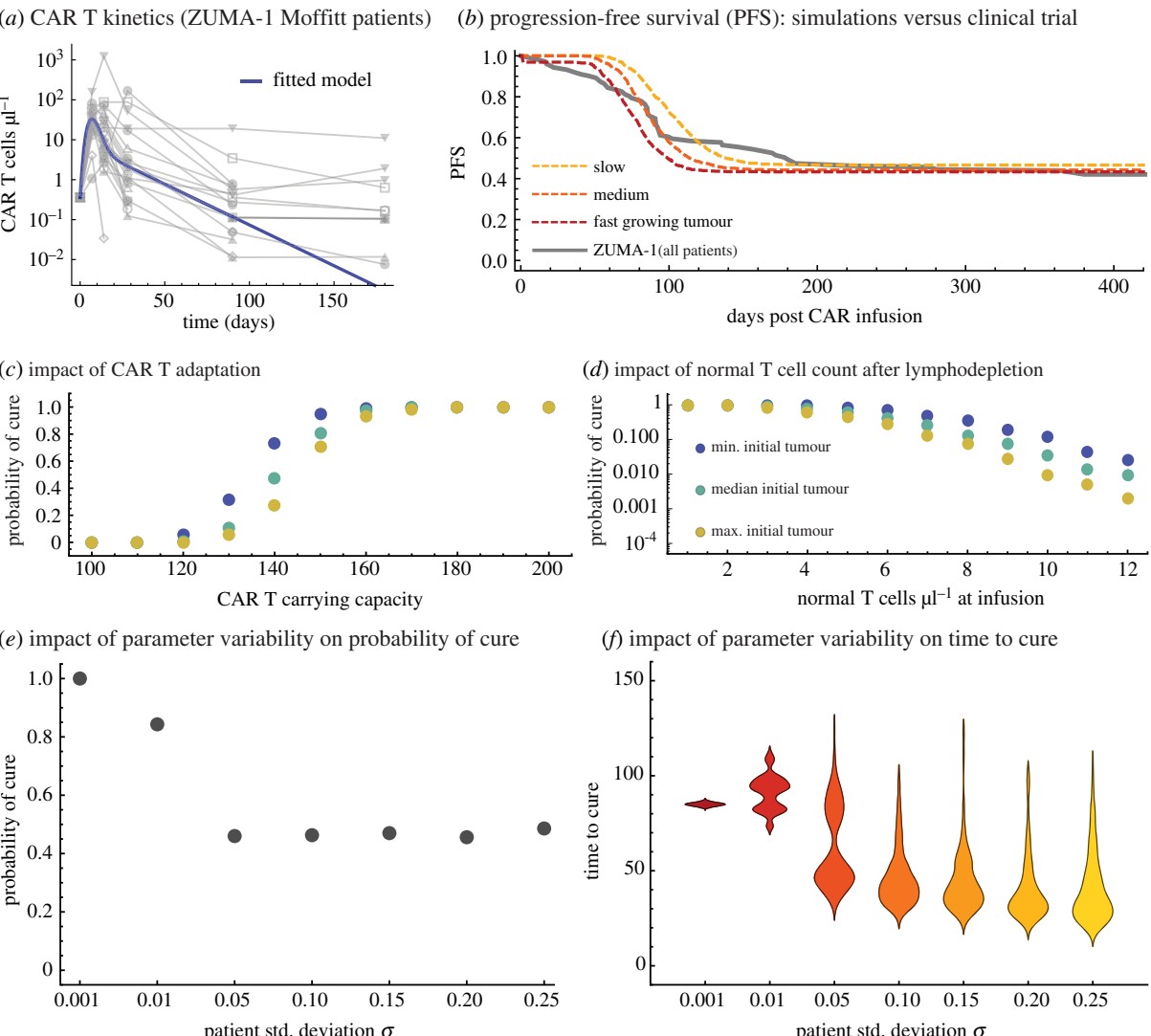

**Figure 3.** Using stochastic simulations to describe progression-free survival and probability of cure (tumour extinction). (*a*) Individual CAR T kinetics for a subset of patients treated on the ZUMA-1 trial (*n* = 21). (*b*) Progression-free survival (PFS) during ZUMA-1 (grey line, *n* = 101), and our stochastic simulations that recapitulates this PFS curve. We recorded progression when 1.2 times the initial tumour mass was reached to avoid bias against cases that briefly increased in tumour mass but responded well eventually. Intrinsic tumour growth rates: 0.105/day (slow), 0.19/day (medium) and 0.265/day (fast). (*c*) Increasing the adaptability of CAR T cells (carrying capacity) improves probability of cure. (*d*) Initial reduction of normal T cells at CAR administration due to lymphodepletion is crucial; increasing this number decreases probability of cure. Tumour sizes for (*c*), (*d*) min = $2.3 \times 10^9$, median = $9.5 \times 10^{10}$, max = $1.3 \times 10^{12}$. All probabilities estimated using 1000 stochastic simulations with the same initial conditions, all other parameters drawn from a normal distribution with a variance of 5% of mean values (table 1). (*e*) Without patient variability (described in the electronic supplementary material, section 5), our set of parameters (table 1) is most likely to lead to cure. As parameter variability increases, cure probability drops and saturates. (*f*) The distribution of time to cure (tumour extinction) without patient variability is narrow. Increases in patient variability then drastically increase the range of possible times to cure. The results in (*e*) and (*f*) were obtained using 100 independent stochastic simulations of the simplified stochastic process described in electronic supplementary material, section 5.4. (Online version in colour.)

parameter sensitivity and correlation analyses, and calculated probability of cure and PFS from stochastic simulations.

We do not make patient-specific, personalized predictions, for which a more comprehensive dataset would be needed [29], matching multiple longitudinal data from the same patient. Instead, we give proof-of-principle that the integration of longitudinal lymphocyte counts with CAR T cell counts and changes in tumour burden can be very useful to predictively model the cell population dynamics that likely determine clinical outcomes.

Our model confirms the hypothesis that sufficient lymphodepletion is an important factor in determining durable response. Improving the adaptation of CAR T cells to expand more and survive longer *in vivo* could result in increased likelihood and duration of response. Future modelling should investigate other available signals, such

as the dynamics and upregulation of homeostatic and inflammatory cytokines.

The emergence of Gompertz growth of T cells is an interesting result of our analysis. A possible explanation can be found in recent work [30], which employed techniques from statistical mechanics to explain the emergence of well-known tumour growth laws. In particular, Gompertz emerged via a reduction in available microstates (e.g. cellular phenotypes), causing a characteristic slowdown of the overall population expansion. Our results suggest that this phenomenon could play a role during immune reconstitution. After rapid expansion, the immune compartment engages in negative selection to keep a flexible adaptive immune system ready to engage pathogens. Therefore, by analogy to the reduction of available microstates, the T cell population approaches a carrying capacity via Gompertz-like growth.

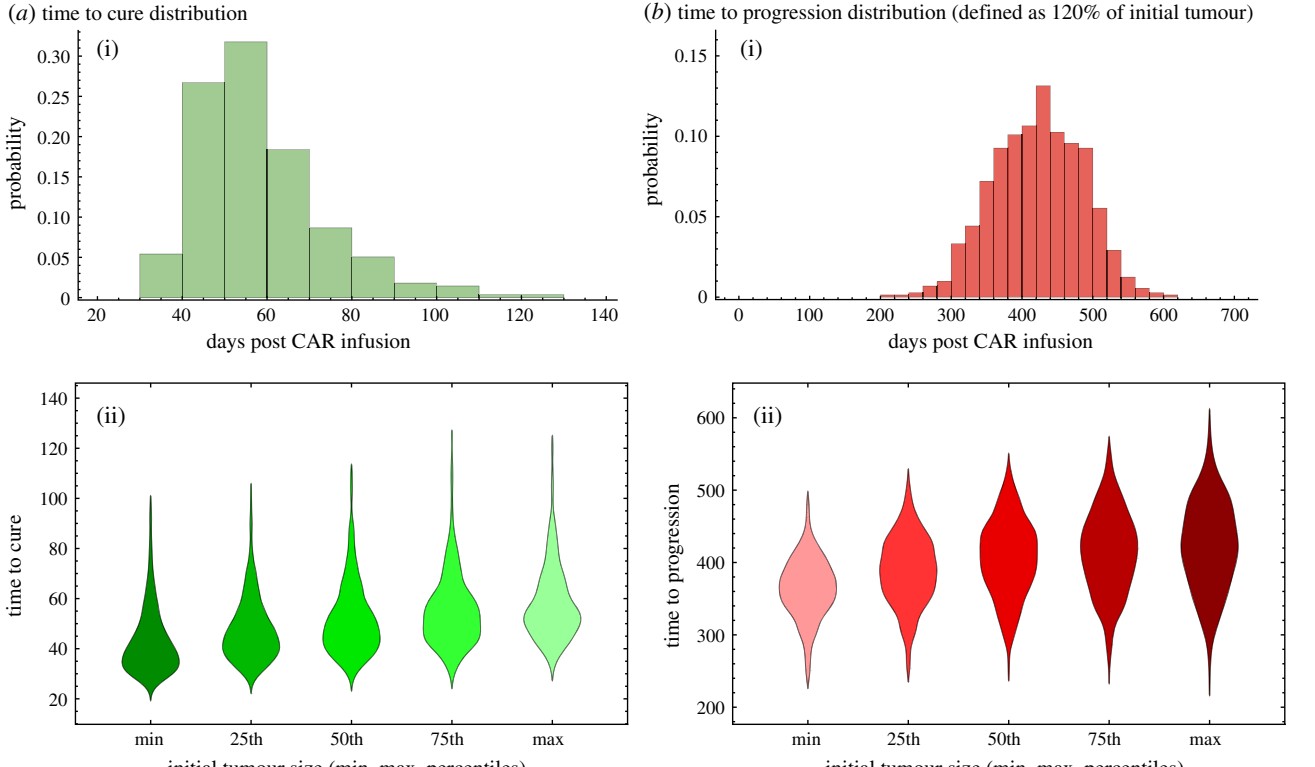

**Figure 4.** Statistics of cure and progression. (*a*) The distribution of cure times for the median parameters. Most patients are cured before day 100. (*b*) The distribution of progression times for the median parameters. Most simulations progress (to 1.2 initial tumour) between days 300 and 500. All parameter values used are given in table 1. All probabilities calculated from 1000 stochastic simulations with the same initial condition. (Online version in colour.)

We made several assumptions to approach the broader biological context of CAR T cell therapy. First, we assumed that immune reconstitution varies minimally across patients. The interplay between normal and CAR T competition in individual patients should be addressed in future studies that track both simultaneously, which could also reveal whether the slope of ALC is predictive of CAR expansion and efficacy.

Second, our model assumes that tumour cell proliferation is independent of tumour burden. However, the tumour growth rate might decrease with tumour burden. In this context, one could explore other sources of tumour burden variability that originate from a logistic dependence of proliferation on tumour volume, called proliferation-saturation [31,32], which points to the need for higher temporal resolution of tumour data.

Third, our probabilistic measure of PFS did not include the evolution of resistance to CAR T cell therapy by immunologic, genetic or epigenetic escape [33], which would add an additional probabilistic modelling layer.

Fourth, normal CD19+ B cells are at negligible levels in the weeks following lymphodepletion. These levels are low until 4–6 months post infusion, implying that their presence is minimal, although they are potential sources of target antigen. Further, the detection of normal CD19+ B cells in circulation long after CAR T is evidence that functional CAR T cells no longer persist in the host. It is unclear whether B cells themselves could be responsible for continued CAR T persistence. Thus, we assume that non-tumour sources of CD19 do not play a role during the activity of CAR T cells.

Fifth, the functional form for the tumour-killing rate was justified in preliminary analysis and by considering the fact that there should be an upper limit on the number of CAR

T cells that can surround a given tumour cell. We expect an upper limit on the rate of killing. However, an analogous argument could be made for tumour influence, leading to the following alternative forms of the killing term: $\gamma_B B\, C/(k_B + \phi B + C)$ or $\gamma_B(B/k_B + B)(C/k_C + C)$. Given the lack of temporal data (especially for the tumour), we elected the simpler form in equation (2.3).

We focused on the treatment of LBCL, but our approach could also be applied to CAR T cell treatment of chronic lymphocytic leukaemia (CLL) [34,35] (investigational), or acute lymphocytic leukaemia (ALL) [36] and mantle cell lymphoma [37], the other approved indications for CAR T cell therapy. Quantitative systems pharmacological (QSP) modelling of CAR T cell kinetics to treat ALL has been conducted recently to describe kinetics independently of tumour or normal T cell dynamics [6,38], to study the effects of additional prophylactic interventions [6], or to include cytokine kinetics [38]. Our model can be interpreted such that long-term CAR T cell survival, possibly necessary to cure ALL, would require a significant slowdown of CAR T cell turnover, or an additional memory compartment. There is not enough longitudinal data in LBCL at this point to model additional CAR T compartments. As such, our model has markedly fewer parameters (two from literature, seven fitted) than several of the recently developed QSP approaches (around 20 parameters) [11].

Our definition of progression predicts progression events later than those observed in some patients per clinical definition [4,17]. This discrepancy indicates that there is additional, unresolved patient heterogeneity, potentially in the form of differences in naive and memory cells in the CAR T cell product at day 0, further highlighting the need for high-resolution longitudinal data.

Our results point to the importance of the immune system's influence on CAR T cell kinetics, and the impact of those kinetics on potentially stochastic tumour dynamics. We hope that our model can be consolidated with descriptions of short-term changes in inflammatory cytokines [19,39–41], because these are accessible alternative biomarkers for the immune system's impact [11]. On the other hand, engineering a CAR T product with fewer cell divisions, improving its 'stemness' or increasing metabolic capacity of CAR T cells should lead to a higher carrying capacity. Such changes would lead to a higher peak and a higher total volume of CAR T cells during treatment. Stem-ness, support by secreted molecules (e.g. IL-2, IL-12) [42] or CAR T cell exhaustion as additional mechanisms should be subject to future modelling.

Finally, models of treatment–tumour interactions do not require that tumour extinction is a stable steady state. Tumour extinction (cure) can be a stochastic event, since cure becomes an absorbing state in a stochastic framework. Our results indicate that the effects of CAR T cell therapy are transient and should be optimized to maximize initial impact and rapidly drive tumours into this stochastic

regime. The dynamics of our model are most sensitive to the ability of CAR T to kill tumours and to the CAR T cell expansion capacity. Future translational work to improve these parameters may ultimately improve efficacy of therapy.

Data accessibility. All code and data used in this manuscript can be found at https://github.com/MathOnco/CARTecology.

Authors' contributions. G.J.K., F.L.L. and P.M.A. conceived of the model and designed the study. G.J.K. carried out the mathematical modelling. G.J.K. and P.M.A. carried out computational modelling and statistical data analysis. F.L.L. and P.M.A. coordinated and supervised the study. G.J.K., F.L.L. and P.M.A. wrote the manuscript. All authors gave final approval for publication and agree to be held accountable for the work performed therein.

Competing interests. G.J.K. and P.M.A. declare no potential conflict of interest. F.L.L. is scientific adviser to Kite, Novartis, and Gamma-Delta T cell Therapeutics, and consultant to CBMG.

Funding. This study was supported by the Richard O. Jacobson Foundation, Moffitt Cancer Center Evolutionary Therapy Center, William G. 'Bill' Bankhead Jr and David Coley Cancer Research Program (20B06), National Cancer Institute (P30-CA076292 and U54-CA193489) and USAMRAA (KC180036).

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
