## [Peer Review File · Proceedings of the Royal Society B: Biological Sciences]

Review History

RSPB-2020-1301.R0 (Original submission)

Review form: Reviewer 1

Recommendation

Major revision is needed (please make suggestions in comments)

Scientific importance: Is the manuscript an original and important contribution to its field?

Good

General interest: Is the paper of sufficient general interest?

Good

Quality of the paper: Is the overall quality of the paper suitable?

Good

Is the length of the paper justified?

Yes

Should the paper be seen by a specialist statistical reviewer?

No

Do you have any concerns about statistical analyses in this paper? If so, please specify them explicitly in your report.

No

It is a condition of publication that authors make their supporting data, code and materials available - either as supplementary material or hosted in an external repository. Please rate, if applicable, the supporting data on the following criteria.

Is it accessible?

Yes

Is it clear?

No

Is it adequate?

No

Do you have any ethical concerns with this paper?

No

Comments to the Author

Major Comments:

1. The role of memory cells is unclear. As described, the model pools naïve and memory populations of normal and CAR T cells. Presumably all subsequent references to memory populations are in fact references to the naïve-memory pooled population.
2. The authors argue that the growth rate of the CAR T cells should decline after some time τ , and suggest that a mechanistic explanation for this property is attack by normal T cells. However, the change in the growth rate is independent of the population of normal T cells. This may not be relevant to the arguments of the paper if this decline in growth rate is adequately supported empirically.
3. The model allows for (and finds through fitting) different carrying capacities for normal and CAR cells. How do the authors interpret this biologically?
4. The measurements used for model fitting appear to be: five time points of lymphocyte counts, six time points of CAR T cell measurements, four time points of tumor size estimates. Median values appear to have been used in all cases, which I interpret to mean that fifteen data points are used to fit the data. The number of parameters fit in the model is sixteen. This raises concerns about the identifiability of the model parameters. As acknowledged in Figure 2 B and 2C, multiple parameterizations fit the data, with a range over τ that seems to cover a range of biologically meaningful outcomes. While the scope of the model has the benefit of proposing mechanistic explanations, a reduced-order model may provide stronger support to the claim. If the authors' key claim is that the timing (τ) of decline in CAR T expansion rate predicts outcomes, then the simulations can provide a plausible mechanistic basis and plausible parameter ranges, but a phenomenological model fitting only τ might be sturdier.

Review form: Reviewer 2

Recommendation

Reject – article is not of sufficient interest (we will consider a transfer to another journal)

Scientific importance: Is the manuscript an original and important contribution to its field?

Acceptable

General interest: Is the paper of sufficient general interest?

Good

Quality of the paper: Is the overall quality of the paper suitable?

Marginal

Is the length of the paper justified?

No

Should the paper be seen by a specialist statistical reviewer?

No

Do you have any concerns about statistical analyses in this paper? If so, please specify them explicitly in your report.

Yes

It is a condition of publication that authors make their supporting data, code and materials available - either as supplementary material or hosted in an external repository. Please rate, if applicable, the supporting data on the following criteria.

Is it accessible?

Yes

Is it clear?

No

Is it adequate?

No

Do you have any ethical concerns with this paper?

No

Comments to the Author

This paper presents an appealingly simple model of CAR T cell dynamics, and links that model with data to propose that early suppression of a tumor could lead to eradication through stochastic fluctuations or to later tumor resurgence.

I have two main problems with the paper as presented. First, the presentation itself is extremely difficult to follow, and second the model includes non-autonomous terms that for me call into question the importance of the mechanisms.

I haven't attempted to point out all the places where the presentation is difficult to follow, but think that a clear description of the data available, followed by a far shorter and more direct exposition of the deterministic model would set things up to present the key results and motivate the stochastic model. I found myself wondering why the paper seems so complex when the strength of the approach is precisely the opposite. I wish the authors would work to put as much of the paper into the main text and reserve the supplement for material, like details of parameter estimation, that are not integral to the paper. The Results on page 9, for example, are loaded with references to supplementary figures. As I said above, if this paper were presented more directly, it could significantly shorter and more focused, and be much better appreciated. For example, Figure 1 does a nice job of laying out the structure and could be used to set things up.

The non-autonomous terms on page 3 are introduced as being "of note", but with no motivation for not making them part of an autonomous model. If indeed the mechanism is attack by "wildtype" T cells (a term not used before in the paper) then why not include that autonomous

mechanism? As an extreme version of this, although of course not what this paper does, one could say that a state variable follow the differential equation $dx/dt = f(t)$ and find an $f(t)$ which gives a great fit to the data. But $x(t) = F(t)$ where F is the integral of f , and we have done nothing more than fit the data. Models like this are about mechanisms. At the bottom of page 6 there is a key point about timing that I think the paper should be focused around directly, with explicit modeling of the immunogenic mechanism and ideally some alternatives.

I am not sufficiently familiar with these treatments to say, but I found the emphasis on using the model to test the effects of a second dose rather poorly supported, particularly if we do not understand the mechanism leading to the apparently immunogenicity. The scenarios at the end of page 10 seem kind of a stretch, and I do not think this application is necessary, particularly if removing this would help to reorganize the paper.

At the end, the more I thought about this paper, the more unclear it became what the critical predictions are. Any model where a tumor or population in general is rapidly reduced and then rebounds can be explored in this way. What is different about CAR T cells from any other mechanism, and what do we learn from the modeling that we didn't know in advance?

I apologize for not having more clear direction in reorganizing this paper, and hope that these comments provide motivation to present this very interesting work in a way that gets it the attention it deserves.

More minor comments:

Abstract:

I find it confusing when CAR is used without the "T". I'm not sure if this was a mistake or a misunderstanding by me.

There are many terms used without being introduced: "immune reconstitution dynamics" and immunogenicity are examples.

This problem is even more severe in the Introduction as noted below.

"rate-disadvantage": Pretty sure the hyphen is in the wrong spot.

Page 2:

Overall response and complete response are not defined, and I can't see how these numbers are consistent with the 60% at the end of the first paragraph.

Is the mixed effect modeling statistical modeling?

At the end of the page, "homeostatic niche" is not defined, and it is not yet clear what feeds back to CAR T cell differentiation.

Page 3:

It would be good to provide more justification for the choice of these particular four state variables.

The use of a form that I think leads to Gompertz growth is justified to some extent in the supplement. Something about this should definitely be included here.

All the material on CCR7 comes completely out of the blue! What does this mean?

Why choose the min functional form in the equation for r_E ?

Why is it "unlikely" for $K_m > K_n$? (also, I think the capitalization of the subscripts is inconsistent).

Page 6:

I don't think papers need to say that things are "novel" -- the results should speak for themselves. And there is a long tradition of immunological modeling that doesn't look fundamentally different from this, which is not at all a bad thing. I did a quick search and found the following for CAR T therapy. They might not be very appropriate, but there is a lot of literature in this area.

```
@article{mostolizadeh2018mathematical,
  title={Mathematical model of chimeric anti-gene receptor (CAR) T cell therapy with presence of cytokine},
  author={Mostolizadeh, Reihaneh and Afsharnezhad, Zahra and Marciniak-Czochra, Anna},
  journal={Numerical Algebra, Control \& Optimization},
  volume={8},
  number={1},
  pages={63},
  year={2018},
```

```
@article{hardiansyah2019quantitative,
  title={Quantitative Systems Pharmacology Model of Chimeric Antigen Receptor T-Cell Therapy},
  author={Hardiansyah, Deni and Ng, Chee Meng},
  journal={Clinical and translational science},
  volume={12},
  number={4},
  pages={343--349},
  year={2019},
```

The second half of the first paragraph seems out of place, and should be integrated into the flow of the paper.

Throughout, I found the use of the term co-evolutionary distracting. What we have is coupled dynamics, with no evolution in the narrow sense.

The parameter estimates in the middle paragraph are not given with any citations.

Page 7:

I don't think this is an extinction vortex -- just extinction due to demographic stochasticity without the addition factors that characterize the vortex, such as increased variability or inbreeding.

The sentence starting "In particular" seems garbled and is definitely awkward.`

Figure 3:

I was confused by the labels of panels D and E as "Survival" in the title and "Probability of cure" on the vertical axis.

Page 9:

Are there any data to predict the degree of patient variability? I would imagine it could be quite large.

Page 11:

The Discussion jumps right into the middle, and should begin with a summary of the key questions and findings, including defining what patient outcomes are being discussed. The problems with terms that have not been introduced or motivated reappears here.

Page 12:

Many paragraphs end with generic statements about the value of the modeling approach, or that more modeling work is needed in the future.

For example, why integrate cytokines? What does the model miss?

As a detail, I thought that IL-7 and IL-15 were less inflammatory and more immunoregulatory.

Page 13:

I was a bit confused by the statement that cures "way past 100 days" (which is a bit too casual a term) are driven by some other mechanism.

This puts a lot of faith in the model getting it right for the most common mechanisms, which I don't see being justified at this point in our understanding.

The final paragraph loads up a lot of immunological detail, which seems out of place. The paper ends with a rather generic statement that models are useful. What have we learned and what specific questions has the modeling opened up?

Page 15:

Not sure if this is a typo or if I didn't understand, but the lines on N and M+E seem inconsistent ($3.0e9/6 = 5e8$ and $1.8e9/0.36=5e9$).

Supplementary Material:

(I apologize for any repeated points, but I typed these up at different times.

The first paragraph of the Introduction is almost impossible to follow.

Why use the median? Are the data time series on individual patients or do only have aggregated data?

As just one example of how confusing the writing is "these counts include CAR T cell density, yet differences in parameters estimates were minimal and no changes in homeostatic ALC were detected" Different in what?

The references and powers on page 2 on not properly formatted.

This was very hard to follow: "Patients in none of the categories (CR, SD, PD) after 1000 days would be defined as undetermined, which did not occur in simulations."

How realistic is it to make T cell populations deterministic?

Page 4: Variance and mean have different units, so I don't see how the variance could be 5-15% of the mean.

Page 4: "We can see by plugging in a few choices for θ that the impact of changing the threshold is minimal". This is kind of hard to evaluate.

This section is kind of wordy also. If the point is that with an exponentially growing tumor different thresholds occur at similar times, say that. And is that level of growth realistic? The one day difference cited here seems rather extreme.

I'm not convinced that section 3 is needed. If we saw an explicit fit to the data that worked well for some aspects of the data and not for others, that would help understand the system, but this seems like an exercise for the authors.

Section 4 opens with an unmotivated statement about non-autonomous behavior. As I say in my main comments, such an assumption begs the question, because nothing in biology is a function of time, but is a function of other state variables.

Page 5: "Third, the CAR T cell products volume (population size) and composition should be integrated, leading to a CAR T cell population that is sub-divided into at least two populations, for example (central) memory and effector CAR T cells." I cannot understand the logical link.

Page 6: As I also mention in the main section, the terminology of co-evolution is very confusing to people who work in ecology, and I cannot see what is game theoretic about a model of population dynamics.

Supp Fig 1: Heterogeneity among patients can cause the appearance of multiple decay rates. Is it possible to analyze data in this way, or, if not, compare models that assume a changing rate with those that assume heterogeneity of rates?

Page 7: This section is quite wordy and hard to follow. I don't see the system as being that complicated, which is a virtue.

Page 8: It would be good to see some indication of the preliminary analysis arguing that β is small and r_i is large. I'm not sure what large means for a parameter with units.

Page 9: I can't understand why there is no coupling of the memory compartments to anything. For example, there is no loss term when memory cell become effector cells. And why would K_N and K_M be different?

Page 10: I don't think the conditions for a peak need to be presented.

Page 10: "some of these patients" is rather vague.

Page 10: I can't see how assuming a fixed fraction of CAR T cells is consistent with the model.

Page 11: Why would CAR T cells numbers be normally distributed? Is this a model of stochastic error, process error or measurement error?

Page 11: In my experience, these models can be sensitive to estimates of carrying capacity. Are results robust if K_N is not 500?

Supp Fig 2: It is hard to justify, or check, introduction of a whole new parameter to fit a single day point.

Supp Fig 3: Would it be possible to show how parameter estimates covary?

Review form: Reviewer 3

Recommendation

Major revision is needed (please make suggestions in comments)

Scientific importance: Is the manuscript an original and important contribution to its field?

Good

General interest: Is the paper of sufficient general interest?

Good

Quality of the paper: Is the overall quality of the paper suitable?

Acceptable

Is the length of the paper justified?

Yes

Should the paper be seen by a specialist statistical reviewer?

No

Do you have any concerns about statistical analyses in this paper? If so, please specify them explicitly in your report.

No

It is a condition of publication that authors make their supporting data, code and materials available - either as supplementary material or hosted in an external repository. Please rate, if applicable, the supporting data on the following criteria.

Is it accessible?

No

Is it clear?

Yes

Is it adequate?

Yes

Do you have any ethical concerns with this paper?

No

Comments to the Author

General interest:

Dynamic properties of CAR T cells have poorly been approached by mathematical modelling so far, and this article is linked with very important questions linked with this new therapy. The model has the potential to predict patient heterogeneity and to be used to optimize decisions.

Summary:

In this manuscript, the authors propose an ODE-based mathematical model to account for the dynamics of injected CAR T cells in patients with B cell tumors (LBCS) after an initial lymphodepletion treatment. The model is further simulated as stochastic birth and death process to estimate the stochastic variation between complete clearance or relapse times.

The model contains four main compartments: The growing tumor, the growing population of endogenous T cells after the lymphodepletion, the injected CAR T cells as memory pool, and finally the effector CAR T cells coming from the differentiation of the memory CAR T cells. The ODE model is fitted on the dynamics of normal T cells, and CAR T cells in a clinical trial for anti CD19 CAR T cells, i.e. T cells targeting killing B cells, including the LBCL tumor. The main experimental property is the fast decay of CAR T cells around 20 days, for unknown reasons. The model explains this decay by assuming a direct negative feedback from normal T cells to CAR T cells using a logistic growth/carrying capacity shared between both cell types, assuming that normal T cells with higher affinity will expand with time as compared with CAR T cells (called 'immunogenicity' here). Accordingly, the fitting explains the CAR T cells dynamics by a 10-fold reduction of their growth rate $r(t)$ due to expanding T cells, sharply around 19 days (Figure 2).

By stochastic simulations, they identify a time-zone where the tumor either completely clears or relapses. They show that higher levels of CAR T cells injection in the beginning increase the probability of cure, but independently on the tumor growth rate. They recapitulate the survival curve under a certain range of tumor growth. Using a gaussian sampling around the best fitted parameters, the model predicts a large variation of the time to relapse (Figure 3).

They predict the effect of a second injection of CAR T cells (Figure 4). Since at later points, the normal T cells strongly inhibit the growth of CAR T cells, a late injection of CAR T cell is predicted to have very minor effect. The authors propose to apply a second treatment in the early phase, to avoid the 'immunogenicity' of reconstituting T cells, by delaying the 10-fold inhibition of CAR T cells growth rate from 19 to 38 days (Figure 4), in which case a second injection of CAR T cells has a positive effect when administered before 40 days, although the initial wave of CAR T cells might still be present then, so it's not clear whether the second wave is the reason of the better outcome.

The authors claim that: - they identified that immunogenicity happens at day 19, - their model confirms the hypothesis that immunogenicity is an important factor for the CAR T cells to disappear

Major points:

1: In order to claim that 'immunogenicity' is the reason for CAR T cell decay, they should provide alternative hypotheses and show that immunogenicity is most consistent with the dataset. Although immunogenicity is a possible mechanism, the authors didn't support adequately this hypothesis from the literature, therefore I am not 100% convinced that this is the reason.

Alternative hypotheses:

- o Activated T cells produce IL2 which is a positive feedback for other T cells, so they could actually support each-other, and the limiting factor could be that it is hard to access the tumor (especially if the collagen network is impacted in the zones of the tumor). CAR T cell decay could be that they are exhausted or do not manage to access the tumor cells.

- o It is not clear how the inject CAR T cell memory pool can sustain the production of effector T cells. Do they have a maximum number of divisions? Do all memory cells become effector with a short life or all becoming exhausted? These hypotheses could explained a synchronized shut down of CAR T cell numbers without needing an 'immunogenicity' hypothesis.

Therefore, the authors should decide whether they want to keep the claim that immunogenicity is the reason (and support it), or remove this claim.

Note: CAR T cells are supposed to have already high affinity... has anybody shown that late T cells have actually higher affinity than the CAR T cells?

2. I have a major concern regarding the fitting of the experimental dataset. Not only only the 'average patient' is taken, but the number of model parameter is bigger than the number of data-points, which suggests overfitting. Indeed, Figure 2C, the time of 'immunogenicity switch' shows a bimodal distribution while this is the same fitting repeated multiple times.

=> Could the authors provide a more in-depth description of the parameter uncertainty [for instance they do not show the distribution of other fitted parameters]. Are some parameters actually identifiable?

=> How does the bimodal distribution impacts on the predictions of relapse times? I want to make sure that the large predictions are not just a consequence of highly different fitting parameters at each fitting.

=> I could not understand whether the authors keep a population of parameter sets from the fitting to make predictions, or whether they just take the parameter set shown in Table 1. In the latter case, are there other similarly optimal parameter sets? Do they predict the same?

3. The predictions on the stochasticity of relapse times is based on gaussian distribution around the optimal parameter set (fitted only to the 'median patient'), which doesn't seem to be accounting for patient heterogeneity. [although the detail how they did is well explained in the ESM]. Why not performing a bootstrap on the full dataset at least, to estimate the population variation?

4. The authors predict a large variation in the time to relapse, which is pretty interesting. Could the authors predict the best time-points to measure relapse (like design a schedule that doctors would need to follow). I suggest to look at this work, <https://doi.org/10.1371/journal.ppat.1005535>, that predicts the best time-points to detect HIV relapse.

5. I am still not convinced that immunogenicity is the reason for CAR T cell decay, but I think this model can still be useful [and that other mechanism could actually maybe lead to similar equations]. Could the authors predict an experimental setting that would actually answer whether immunogenicity is the reason?

6. The model used for the immune interactions is partially analyzed in the ESM but are not linked from the main manuscript, and still I think more analyses of the model property could be beneficial. For instance, do the mutual inhibition interaction motifs confer hysteresis and tristability? A more in-depth analysis of the dynamics and the mechanistic consequences of the model design would give a better understanding of the properties of this model.

7a. The proposition to re-inject CAR T cell is interesting. But since the authors keep the same immunogenicity of normal T cells, the predictions for Figure 4 are pretty straightforward and it is not very helpful to reinject CAR T cells after 20 days. Is there experimental back up to state that reinjection of CAR T cells has no effect on the PFS? It is hard to believe this statement. Now the authors propose to add 2 more steps: reducing the immunogenicity and then injecting CAR T cells again. This sounds pretty heavy for the patient. Could the authors discuss or use the model to propose a different (one step only) therapy? For instance, they didn't talk about exhaustion or check-point. Could they speculate whether checkpoint inhibition or drugs modulating the metabolism of T cells could differentially help CAR T cells versus normal T cells?

7b. Could the authors show whether, for $r_2 = 37.6$ days, the long-term FPS improvement is due to the first prolonged wave of CAR T cells, or whether the second one is actually the reason for it?

8. In the discussion: «We identified 5 processes as potential drivers of patient outcomes». The authors just mention the mechanisms in the model, but actually do not investigate the effect of these mechanisms on the outcome. Either investigate or do not say they are identified as drivers...

Minor points:

There are interesting analyses in the supplementary file, but they are poorly related from the main manuscript.

Please provide line numbers in the next manuscript!

Too strong statement in the abstract: The model demonstrate that CAR expansion is shaped by immune reconstitution. You do not prove that (see point 1)

Can you explain why only N (normal T cells) and M (CAR memory) share the carrying capacity, but not the effector pool?

Figure 2, why not plotting the curves for both M(t) and E(t)?

Table 1: Please give the standard deviation of the fitted parameters.

The authors state that normal T cells replace CAR T cells because they have higher affinity, fine, but then why would the CAR T cells be more efficient at killing the tumor than higher affinity normal T cells? For instance, page 13 they say «Further, detection of normal CD19+ B cells long after CAR T is likely evidence that functional CAR T cells no longer persist in the host»

Discussion:

The authors should discuss to which extent their study is proper to LBCL and whether it actually applies to other CAR T cell therapies. For instance, they start after lymphodepletion.

Decision letter (RSPB-2020-1301.R0)

03-Aug-2020

Dear Dr Altrock:

I am writing to inform you that your manuscript RSPB-2020-1301 entitled "Response to CAR T cell therapy can be explained by co-evolutionary cell dynamics and stochastic extinction events" has, in its current form, been rejected for publication in Proceedings B.

This action has been taken on the advice of referees, who have recommended that substantial revisions are necessary. With this in mind we would be happy to consider a resubmission, provided the comments of the referees are fully addressed. However please note that this is not a provisional acceptance.

Sincerely,

Dr Sasha Dall

Associate Editor

Comments to Author:

The reviewers make extensive and significant points about the scientific content of the paper, and the presentation. The reviewers also emphasize (and I agree) the recommendation that the paper should be thoroughly re-written, almost from the ground up, in order to reach its full potential. A very concerning point surrounds the parameter fitting - where it does appear that the fit models have more parameters than the number of data points. Clarity around the fitting process here will be essential.

Reviewer(s)' Comments to Author:

Referee: 1

Comments to the Author(s)

Major Comments:

1. The role of memory cells is unclear. As described, the model pools naïve and memory populations of normal and CAR T cells. Presumably all subsequent references to memory populations are in fact references to the naïve-memory pooled population.
2. The authors argue that the growth rate of the CAR T cells should decline after some time τ , and suggest that a mechanistic explanation for this property is attack by normal T cells. However, the change in the growth rate is independent of the population of normal T cells. This may not be relevant to the arguments of the paper if this decline in growth rate is adequately supported empirically.
3. The model allows for (and finds through fitting) different carrying capacities for normal and CAR cells. How do the authors interpret this biologically?
4. The measurements used for model fitting appear to be: five time points of lymphocyte counts, six time points of CAR T cell measurements, four time points of tumor size estimates. Median values appear to have been used in all cases, which I interpret to mean that fifteen data points are used to fit the data. The number of parameters fit in the model is sixteen. This raises concerns about the identifiability of the model parameters. As acknowledged in Figure 2 B and 2C, multiple parameterizations fit the data, with a range over τ that seems to cover a range of biologically meaningful outcomes. While the scope of the model has the benefit of proposing mechanistic explanations, a reduced-order model may provide stronger support to the claim. If the authors' key claim is that the timing (τ) of decline in CAR T expansion rate predicts outcomes, then the simulations can provide a plausible mechanistic basis and plausible parameter ranges, but a phenomenological model fitting only τ might be sturdier.

Referee: 2

Comments to the Author(s)

This paper presents an appealingly simple model of CAR T cell dynamics, and links that model with data to propose that early suppression of a tumor could lead to eradication through stochastic fluctuations or to later tumor resurgence.

I have two main problems with the paper as presented. First, the presentation itself is extremely difficult to follow, and second the model includes non-autonomous terms that for me call into question the importance of the mechanisms.

I haven't attempted to point out all the places where the presentation is difficult to follow, but think that a clear description of the data available, followed by a far shorter and more direct exposition of the deterministic model would set things up to present the key results and motivate the stochastic model. I found myself wondering why the paper seems so complex when the strength of the approach is precisely the opposite. I wish the authors would work to put as much of the paper into the main text and reserve the supplement for material, like details of parameter estimation, that are not integral to the paper. The Results on page 9, for example, are loaded with references to supplementary figures. As I said above, if this paper were presented more directly,

it could significantly shorter and more focused, and be much better appreciated. For example, Figure 1 does a nice job of laying out the structure and could be used to set things up.

The non-autonomous terms on page 3 are introduced as being "of note", but with no motivation for not making them part of an autonomous model. If indeed the mechanism is attack by "wildtype" T cells (a term not used before in the paper) then why not include that autonomous mechanism? As an extreme version of this, although of course not what this paper does, one could say that a state variable follow the differential equation $dx/dt = f(t)$ and find an $f(t)$ which gives a great fit to the data. But $x(t) = F(t)$ where F is the integral of f , and we have done nothing more than fit the data. Models like this are about mechanisms. At the bottom of page 6 there is a key point about timing that I think the paper should be focused around directly, with explicit modeling of the immunogenic mechanism and ideally some alternatives.

I am not sufficiently familiar with these treatments to say, but I found the emphasis on using the model to test the effects of a second dose rather poorly supported, particularly if we do not understand the mechanism leading to the apparently immunogenicity. The scenarios at the end of page 10 seem kind of a stretch, and I do not think this application is necessary, particularly if removing this would help to reorganize the paper.

At the end, the more I thought about this paper, the more unclear it became what the critical predictions are. Any model where a tumor or population in general is rapidly reduced and then rebounds can be explored in this way. What is different about CAR T cells from any other mechanism, and what do we learn from the modeling that we didn't know in advance?

I apologize for not having more clear direction in reorganizing this paper, and hope that these comments provide motivation to present this very interesting work in a way that gets it the attention it deserves.

More minor comments:

Abstract:

I find it confusing when CAR is used without the "T". I'm not sure if this was a mistake or a misunderstanding by me.

There are many terms used without being introduced: "immune reconstitution dynamics" and immunogenicity are examples. This problem is even more severe in the Introduction as noted below.

"rate-disadvantage": Pretty sure the hyphen is in the wrong spot.

Page 2:

Overall response and complete response are not defined, and I can't see how these numbers are consistent with the 60% at the end of the first paragraph.

Is the mixed effect modeling statistical modeling?

At the end of the page, "homeostatic niche" is not defined, and it is not yet clear what feeds back to CAR T cell differentiation.

Page 3:

It would be good to provide more justification for the choice of these particular four state variables.

The use of a form that I think leads to Gompertz growth is justified to some extent in the supplement. Something about this should definitely be included here.`

All the material on CCR7 comes completely out of the blue! What does this mean?

Why choose the min functional form in the equation for r_E ?

Why is it "unlikely" for $K_m > K_n$? (also, I think the capitalization of the subscripts is inconsistent).

Page 6:

I don't think papers need to say that things are "novel" -- the results should speak for themselves. And there is a long tradition of immunological modeling that doesn't look fundamentally different from this, which is not at all a bad thing. I did a quick search and found the following for CAR T therapy. They might not be very appropriate, but there is a lot of literature in this area.

```
@article{mostolizadeh2018mathematical,
  title={Mathematical model of chimeric anti-gene receptor (CAR) T cell therapy with presence of cytokine},
  author={Mostolizadeh, Reihaneh and Afsharnezhad, Zahra and Marciniak-Czochra, Anna},
  journal={Numerical Algebra, Control \& Optimization},
  volume={8},
  number={1},
  pages={63},
  year={2018},
```

```
@article{hardiansyah2019quantitative,
  title={Quantitative Systems Pharmacology Model of Chimeric Antigen Receptor T-Cell Therapy},
  author={Hardiansyah, Deni and Ng, Chee Meng},
  journal={Clinical and translational science},
  volume={12},
  number={4},
  pages={343--349},
  year={2019},
```

The second half of the first paragraph seems out of place, and should be integrated into the flow of the paper.

Throughout, I found the use of the term co-evolutionary distracting. What we have is coupled dynamics, with no evolution in the narrow sense.

The parameter estimates in the middle paragraph are not given with any citations.

Page 7:

I don't think this is an extinction vortex -- just extinction due to demographic stochasticity without the addition factors that characterize the vortex, such as increased variability or inbreeding.

The sentence starting "In particular" seems garbled and is definitely awkward.`

Figure 3:

I was confused by the labels of panels D and E as "Survival" in the title and "Probability of cure" on the vertical axis.

Page 9:

Are there any data to predict the degree of patient variability? I would imagine it could be quite large.

Page 11:

The Discussion jumps right into the middle, and should begin with a summary of the key questions and findings, including defining what patient outcomes are being discussed. The problems with terms that have not been introduced or motivated reappears here.

Page 12:

Many paragraphs end with generic statements about the value of the modeling approach, or that more modeling work is needed in the future. For example, why integrate cytokines? What does the model miss?

As a detail, I thought that IL-7 and IL-15 were less inflammatory and more immunoregulatory.

Page 13:

I was a bit confused by the statement that cures "way past 100 days" (which is a bit too casual a term) are driven by some other mechanism. This puts a lot of faith in the model getting it right for the most common mechanisms, which I don't see being justified at this point in our understanding.

The final paragraph loads up a lot of immunological detail, which seems out of place. The paper ends with a rather generic statement that models are useful. What have we learned and what specific questions has the modeling opened up?

Page 15:

Not sure if this is a typo or if I didn't understand, but the lines on N and M+E seem inconsistent ($3.0e9/6 = 5e8$ and $1.8e9/0.36=5e9$).

Supplementary Material:

(I apologize for any repeated points, but I typed these up at different times.

The first paragraph of the Introduction is almost impossible to follow.

Why use the median? Are the data time series on individual patients or do only have aggregated data?

As just one example of how confusing the writing is "these counts include CAR T cell density, yet differences in parameters estimates were minimal and no changes in homeostatic ALC were detected" Different in what?

The references and powers on page 2 on not properly formatted.

This was very hard to follow: "Patients in none of the categories (CR, SD, PD) after 1000 days would be defined as undetermined, which did not occur in simulations."

How realistic is it to make T cell populations deterministic?

Page 4: Variance and mean have different units, so I don't see how the variance could be 5-15% of the mean.

Page 4: "We can see by plugging in a few choices for θ that the impact of changing the threshold is minimal". This is kind of hard to evaluate. This section is kind of wordy also. If the point is that with an exponentially growing tumor different thresholds occur at similar times, say that. And is that level of growth realistic? The one day difference cited here seems rather extreme.

I'm not convinced that section 3 is needed. If we saw an explicit fit to the data that worked well for some aspects of the data and not for others, that would help understand the system, but this seems like an exercise for the authors.

Section 4 opens with an unmotivated statement about non-autonomous behavior. As I say in my main comments, such an assumption begs the question, because nothing in biology is a function of time, but is a function of other state variables.

Page 5: "Third, the CAR T cell products volume (population size) and composition should be integrated, leading to a CAR T cell population that is sub-divided into at least two populations, for example (central) memory and effector CAR T cells." I cannot understand the logical link.

Page 6: As I also mention in the main section, the terminology of co-evolution is very confusing to people who work in ecology, and I cannot see what is game theoretic about a model of population dynamics.

Supp Fig 1: Heterogeneity among patients can cause the appearance of multiple decay rates. Is it possible to analyze data in this way, or, if not, compare models that assume a changing rate with those that assume heterogeneity of rates? Page 7: This section is quite wordy and hard to follow. I don't see the system as being that complicated, which is a virtue.

Page 8: It would be good to see some indication of the preliminary analysis arguing that β is small and r_i is large. I'm not sure what large means for a parameter with units.

Page 9: I can't understand why there is no coupling of the memory compartments to anything. For example, there is no loss term when memory cell become effector cells. And why would K_N and K_M be different?

Page 10: I don't think the conditions for a peak need to be presented.

Page 10: "some of these patients" is rather vague.

Page 10: I can't see how assuming a fixed fraction of CAR T cells is consistent with the model.

Page 11: Why would CAR T cells numbers be normally distributed? Is this a model of stochastic error, process error or measurement error?

Page 11: In my experience, these models can be sensitive to estimates of carrying capacity. Are results robust if K_N is not 500?

Supp Fig 2: It is hard to justify, or check, introduction of a whole new parameter to fit a single day point.

Supp Fig 3: Would it be possible to show how parameter estimates covary?

Referee: 3

Comments to the Author(s)

General interest:

Dynamic properties of CAR T cells have poorly been approached by mathematical modelling so far, and this article is linked with very important questions linked with this new therapy. The model has the potential to predict patient heterogeneity and to be used to optimize decisions.

Summary:

In this manuscript, the authors propose an ODE-based mathematical model to account for the dynamics of injected CAR T cells in patients with B cell tumors (LBCL) after an initial lymphodepletion treatment. The model is further simulated as stochastic birth and death process to estimate the stochastic variation between complete clearance or relapse times.

The model contains four main compartments: The growing tumor, the growing population of endogenous T cells after the lymphodepletion, the injected CAR T cells as memory pool, and finally the effector CAR T cells coming from the differentiation of the memory CAR T cells. The ODE model is fitted on the dynamics of normal T cells, and CAR T cells in a clinical trial for anti CD19 CAR T cells, i.e. T cells targeting killing B cells, including the LBCL tumor. The main experimental property is the fast decay of CAR T cells around 20 days, for unknown reasons. The model explains this decay by assuming a direct negative feedback from normal T cells to CAR T cells using a logistic growth/carrying capacity shared between both cell types, assuming that normal T cells with higher affinity will expand with time as compared with CAR T cells (called 'immunogenicity' here). Accordingly, the fitting explains the CAR T cells dynamics by a 10-fold reduction of their growth rate $r(t)$ due to expanding T cells, sharply around 19 days (Figure 2).

By stochastic simulations, they identify a time-zone where the tumor either completely clears or relapses. They show that higher levels of CAR T cells injection in the beginning increase the probability of cure, but independently on the tumor growth rate. They recapitulate the survival curve under a certain range of tumor growth. Using a gaussian sampling around the best fitted parameters, the model predicts a large variation of the time to relapse (Figure 3).

They predict the effect of a second injection of CAR T cells (Figure 4). Since at later points, the normal T cells strongly inhibit the growth of CAR T cells, a late injection of CAR T cell is predicted to have very minor effect. The authors propose to apply a second treatment in the early phase, to avoid the 'immunogenicity' of reconstituting T cells, by delaying the 10-fold inhibition of CAR T cells growth rate from 19 to 38 days (Figure 4), in which case a second injection of CAR T cells has a positive effect when administered before 40 days, although the initial wave of CAR T cells might still be present then, so it's not clear whether the second wave is the reason of the better outcome.

The authors claim that: - they identified that immunogenicity happens at day 19, - their model confirms the hypothesis that immunogenicity is an important factor for the CAR T cells to disappear

Major points:

1: In order to claim that 'immunogenicity' is the reason for CAR T cell decay, they should provide alternative hypotheses and show that immunogenicity is most consistent with the dataset. Although immunogenicity is a possible mechanism, the authors didn't support adequately this hypothesis from the literature, therefore I am not 100% convinced that this is the reason.

Alternative hypotheses:

- o Activated T cells produce IL2 which is a positive feedback for other T cells, so they could actually support each-other, and the limiting factor could be that it is hard to access the tumor (especially if the collagen network is impacted in the zones of the tumor). CAR T cell decay could be that they are exhausted or do not manage to access the tumor cells.

- o It is not clear how the inject CAR T cell memory pool can sustain the production of effector T cells. Do they have a maximum number of divisions? Do all memory cells become effector with a short life or all becoming exhausted? These hypotheses could explained a synchronized shut down of CAR T cell numbers without needing an 'immunogenicity' hypothesis.

Therefore, the authors should decide whether they want to keep the claim that immunogenicity is the reason (and support it), or remove this claim.

Note: CAR T cells are supposed to have already high affinity... has anybody shown that late T cells have actually higher affinity than the CAR T cells?

2. I have a major concern regarding the fitting of the experimental dataset. Not only only the 'average patient' is taken, but the number of model parameter is bigger than the number of data-points, which suggests overfitting. Indeed, Figure 2C, the time of 'immunogenicity switch' shows a bimodal distribution while this is the same fitting repeated multiple times.

=> Could the authors provide a more in-depth description of the parameter uncertainty [for instance they do not show the distribution of other fitted parameters]. Are some parameters actually identifiable?

=> How does the bimodal distribution impacts on the predictions of relapse times? I want to make sure that the large predictions are not just a consequence of highly different fitting parameters at each fitting.

=> I could not understand whether the authors keep a population of parameter sets from the fitting to make predictions, or whether they just take the parameter set shown in Table 1. In the latter case, are there other similarly optimal parameter sets? Do they predict the same?

3. The predictions on the stochasticity of relapse times is based on gaussian distribution around the optimal parameter set (fitted only to the 'median patient'), which doesn't seem to be accounting for patient heterogeneity. [although the detail how they did is well explained in the ESM]. Why not performing a bootstrap on the full dataset at least, to estimate the population variation?

4. The authors predict a large variation in the time to relapse, which is pretty interesting. Could the authors predict the best time-points to measure relapse (like design a schedule that doctors would need to follow). I suggest to look at this work, <https://doi.org/10.1371/journal.ppat.1005535>, that predicts the best time-points to detect HIV relapse.

5. I am still not convinced that immunogenicity is the reason for CAR T cell decay, but I think this model can still be useful [and that other mechanism could actually maybe lead to similar equations]. Could the authors predict an experimental setting that would actually answer whether immunogenicity is the reason?

6. The model used for the immune interactions is partially analyzed in the ESM but are not linked from the main manuscript, and still I think more analyses of the model property could be beneficial. For instance, do the mutual inhibition interaction motifs confer hysteresis and tristability? A more in-depth analysis of the dynamics and the mechanistic consequences of the model design would give a better understanding of the properties of this model.

7a. The proposition to re-inject CAR T cell is interesting. But since the authors keep the same immunogenicity of normal T cells, the predictions for Figure 4 are pretty straightforward and it is not very helpful to reinject CAR T cells after 20 days. Is there experimental back up to state that reinjection of CAR T cells has no effect on the PFS? It is hard to believe this statement.

Now the authors propose to add 2 more steps: reducing the immunogenicity and then injecting CAR T cells again. This sounds pretty heavy for the patient. Could the authors discuss or use the model to propose a different (one step only) therapy? For instance, they didn't talk about exhaustion or check-point. Could they speculate whether checkpoint inhibition or drugs modulating the metabolism of T cells could differentially help CAR T cells versus normal T cells?

7b. Could the authors show whether, for $t_2 = 37.6$ days, the long-term PFS improvement is due to the first prolonged wave of CAR T cells, or whether the second one is actually the reason for it?

8. In the discussion: «We identified 5 processes as potential drivers of patient outcomes». The authors just mention the mechanisms in the model, but actually do not investigate the effect of these mechanisms on the outcome. Either investigate or do not say they are identified as drivers...

Minor points:

There are interesting analyses in the supplementary file, but they are poorly related from the main manuscript.

Please provide line numbers in the next manuscript!

Too strong statement in the abstract: The model demonstrate that CAR expansion is shaped by immune reconstitution. You do not prove that (see point 1)

Can you explain why only N (normal T cells) and M (CAR memory) share the carrying capacity, but not the effector pool?

Figure 2, why not plotting the curves for both $M(t)$ and $E(t)$?

Table 1: Please give the standard deviation of the fitted parameters.

The authors state that normal T cells replace CAR T cells because they have higher affinity, fine, but then why would the CAR T cells be more efficient at killing the tumor than higher affinity normal T cells? For instance, page 13 they say «Further, detection of normal CD19+ B cells long after CAR T is likely evidence that functional CAR T cells no longer persist in the host»

Discussion:

The authors should discuss to which extend their study is proper to LBCL and whether it actually applies to other CAR T cell therapies. For instance, they start after lymphodepletion.

Author's Response to Decision Letter for (RSPB-2020-1301.R0)

See Appendix A.

RSPB-2020-2765.R0

Review form: Reviewer 2

Recommendation

Major revision is needed (please make suggestions in comments)

Scientific importance: Is the manuscript an original and important contribution to its field?

Good

General interest: Is the paper of sufficient general interest?

Good

Quality of the paper: Is the overall quality of the paper suitable?

Good

Is the length of the paper justified?

Yes

Should the paper be seen by a specialist statistical reviewer?

No

Do you have any concerns about statistical analyses in this paper? If so, please specify them explicitly in your report.

No

It is a condition of publication that authors make their supporting data, code and materials available - either as supplementary material or hosted in an external repository. Please rate, if applicable, the supporting data on the following criteria.

Is it accessible?

Yes

Is it clear?

Yes

Is it adequate?

Yes

Do you have any ethical concerns with this paper?

No

Comments to the Author

I enjoyed reading the revision of this paper, and thank the authors for the extremely hard work they put into improving this paper. The model and the presentation are both much clearer, although I still have some significant questions and suggestions on both. Some of the newly added sections could use a bit more editing for consistency and conciseness, but I didn't mark all of those spots.

As I was reading, I found it odd that the deterministic model and stochastic model are in different units. Why not just do everything in cell numbers? The deterministic model will scale, so the results will be identical, and save readers from making one more translation.

The paper discussed this, but it is hard to get a sense of the diverse trajectories followed by patients either from the medians or the quartiles. Would it be possible to see some examples of actual patient data?

I have two remaining concerns about the model. First, I can't see why the saturating term in the B equation has $kB+C$ rather than $kB+B$ in the denominator. The latter produces an upper limit on the killing rate per CAR T cell. With a small value of C, for example, the per capita killing rate increases linearly without bound in B with the existing form.

Second, the $rC(T)$ term is not well motivated. The whole equation still depends only on T, and thus this just gives a different form of competition. I can't see where this form comes from. It just magically appears in the transition from equation (1b) to (2b) in the supplement. What does "immune reconstitution capacity" mean in Table 1? Why is the Gompertz form with lower carrying capacity not sufficient? Page 4 (lines 22-23) seems to be hinting at something along these lines, but need some justification. In addition, it looks from Figure 2C that the results are not very sensitive to the choice of a and b, so perhaps this whole term could be eliminated or simplified.

The paper seems a bit inconsistent about the alternative sources of CD-19. On page 4, it sound like 50% of patients do have detectable normal B cells in circulation. That does not sound negligible to me.

I am concerned that the model results are most sensitive to the values of the carrying capacities. These are problematic parameters because they have little mechanistic interpretation, and I don't see how they could be manipulated in any way by treatment. Although it is perfectly reasonable, it is hardly surprising that increasing the tumor-killing rate of CAR T cells would improve outcomes.

An additional problem with models based on carrying capacity is their failure to separate birth and death rates, such as are discussed on page

7. These almost certainly could not be identified from this sort of data, but interpretations should be more cautious because the negative birth rates that occur when $T \gg KC$ cannot be treated as death rates.

I found the justification of the Gompertz form for immune system growth a bit confusing. Why would tumor growth laws help us understand immune system growth? I'd shorten this part, and just say that it provided a good fit.

The final paragraph doesn't make a very strong conclusion. That extinction is absorbing in a stochastic model even when it might be unstable in a deterministic model is well known. I'd want to end with something more like "Effects of CAR T are transient, and we therefore need to maximize their initial impact and knock tumor cells into the stochastic regime as quickly as possible. Our models describe these dynamics, and point toward mechanisms that might enhance the effectiveness of this therapy in eradicating tumors."

Minor points:

Abstract, line 19: I'm not sure if this is some technical phrasing, but "cure events" sounds awkward. Why not just "cures"?

Line 21: "progression occurs much later and is widely distributed in time. We parameterized our model with time series of CAR T cell populations and tumor size and..."

Line 25: I don't think we can quantify "why". I'd go with "how".

Page 2:

Line 4: Unclear what "Relapsed and refractory" mean here.

Line 5: I'd say that CD19 is a marker rather than a protein.

Line 6: "LBCL patients who do not respond to chemotherapy have a median overall survival of under 7 months".

Line 12: What does "would have been" mean here? I don't see how this lines up with the 82% and 54% above.

Line 30: "stochastic due to small population size of tumor cells."

Line 31: "explain" is a bit strong, perhaps "match". Perhaps "reveals" could be "predicts". Finally, the sentence starting on line 32 doesn't add much and I think could be deleted.

Figure 1A: "co-evolution" is a strange term for the interaction between N and C. Isn't it just competition?

Page 7, line 5: "disfavored" is awkward.

Page 7, line 6: This is pretty wordy. Competitive exclusion works, so I'd cut this sentence and rewrite the next sentences.

"This CAR T cell disadvantage predicts their decline and eventual extinction. Other mechanisms, like CAR T cell effector memory cells could maintain persistence."

Page 7: line 26. How about "Stochastic tumor extinction may explain observed treatment success rates". This section should be more clear that a deterministic model cannot drive any state variable to zero, and thus the tumor will necessarily bounce back. This paragraph could end

more clearly "brought down to very small cell numbers and subject to stochastic extinction."

Page 8, line 1: I don't see why times would follow a broader distribution based on the previous argument. And lines 11-12 on this page could bear with more discussion.

Figure 3: min, median and max are not defined clearly in the figure caption. In that caption, what is adaptability? I might have missed it, but what is the ALC of the six cells?

Page 10, lines 2-3. I'd delete the new phrase and just end with "However, the tumor growth rate might decrease with tumor burden."

Page 10, line 16: Not sure what "driving event" means.

Page 11, line 10-12: I couldn't follow what is being said here.

Page 11, line 21-22: I'd cut the phrase "with which additional complexity..." And rewrite the next sentence

"Our definition of progression predicts progression events later than those observed in some patients."

Page 11, line 29-32: I found this rather confusing also.

Page 12, line 2: This bit on cytokines seems rather out of place.

Review form: Reviewer 3

Recommendation

Major revision is needed (please make suggestions in comments)

Scientific importance: Is the manuscript an original and important contribution to its field?

Acceptable

General interest: Is the paper of sufficient general interest?

Excellent

Quality of the paper: Is the overall quality of the paper suitable?

Acceptable

Is the length of the paper justified?

Yes

Should the paper be seen by a specialist statistical reviewer?

No

Do you have any concerns about statistical analyses in this paper? If so, please specify them explicitly in your report.

No

It is a condition of publication that authors make their supporting data, code and materials available - either as supplementary material or hosted in an external repository. Please rate, if applicable, the supporting data on the following criteria.

Is it accessible?

Yes

Is it clear?

No

Is it adequate?

N/A

Do you have any ethical concerns with this paper?

No

Comments to the Author

In the original manuscript, the model was presented with 3 potential interests: 1/ discriminating possible immune mechanisms responsible for CAR T cells dynamics in vivo, 2/ interesting mathematical properties of the 3-compartment mathematical model and 3/ prediction of treatment optimization with re-injection of CAR-T cells thanks to the model, or predictions of consequences of the model in term of cure or relapse.

In the first round of review, none of these 3 potential interests were addressed convincingly enough and both reviewers raised technical concerns about the mathematical analysis.

Now, after re-submission, the authors solved the technical concerns by re-drawing a new-simpler model with less parameters, that looks convincing enough on the technical level, and is more beautiful/adequate, especially compared to other extremely complicated models in the literature.

However, the authors did remove the parts related to the 'interesting points': 1/ saying their initial claim for one mechanism was controversial so they removed the mechanistic interpretation, and replaced by another controversial interpretation that the carrying capacity is THE reason without explicitly supporting it (see point below); for point 2/ by saying they disagree that a more deep analytical / mathematical analysis of the model would be beneficial and 3/ by removing the figure related to predicting a new therapy. Also, they discarded the question to assess patient heterogeneity from the data.

So the main statement remaining in this paper from the original version, is 1/ to provide a simple mathematical model to explain the the curves of CAR T cell dynamics versus non-CAR T cells from real patient data when there is B cell depletion before CAR T cell therapy, and (newly added) 2/ to describe stochastically when are the expected times of cure versus relapse although this is based on an 'average patient'. Which is already something adding value to previously published papers (that the authors fairly acknowledge), but lacks a bit the 'meat'.

I want to mention that the work to rewrite the manuscript and keep the track changes is very much appreciated, and the readability was already pretty improved.

I therefore think it would be fair to publish this study but with major substantial modifications.

Major points

1- Assess patient heterogeneity and its effect into the stochastic predictions of cure. I accept that not all data points are available, but there are methods to cope with it, would it be a bootstrap on the variability of each time-point, or the comparison of the few longitudinal available patients. This point is critical, as the prediction relies on stochastic variability. In particular, would any patient have the same chance of relapse, or different parameters inferred from different patients would predict different outcome? Meaning: is stochasticity winning over individualized parameters? This would be a critical point for therapy monitoring.

2- Concluding that the carrying capacity of CAR T cells is different than the one of endogenous T cells would need hypothesis comparison. This is true that the model suggests it is possible to explain the dynamics with two different carrying capacity, but does not exclude that other hypothesis could equally well explain the data (especially as the data has not so many time-points). Please thoroughly discuss other possible hypotheses (as the ones I already raised in the first round of revisions), and replace the strong statement by "In the model, different carrying capacities are needed" or "different carrying capacities are consistent with the observed dynamics". - or find ways to better support this statement.

Minor points:

1- Please show the identifiability (profile likelihood?), and parameter correlation as figure, not only in the text

2- Figure 1D is identical to the original manuscript, please update with the new model

3- Supplementary Figure 1 in the ESM is not visible,

4- Please incorporate better the findings of the ESM into the main manuscript. If not the ESM is fairly unmentioned although it brings valuable insights.

5- Define Gompertzian,

6- The results start at Figure 2C, while Figure 2A is already a result. Please take time to explain all Figure 2 in the beginning of the results (eventually, move this part out of the methods)

Decision letter (RSPB-2020-2765.R0)

07-Dec-2020

I am writing to inform you that this version of your manuscript RSPB-2020-2765 entitled "Response to CAR T cell therapy can be explained by T cell dynamics and stochastic extinction events" has, in its current form, been rejected for publication in Proceedings B.

This action has been taken on the advice of referees, who have recommended that substantial revisions are necessary. With this in mind we would be happy to consider a resubmission, provided the comments of the referees are fully addressed. However please note that this is not a provisional acceptance.

Please find below the comments made by the referees, not including confidential reports to the Editor, which I hope you will find useful.

1) A 'response to referees' document including details of how you have responded to the comments, and the adjustments you have made.

- 2) A clean copy of the manuscript and one with 'tracked changes' indicating your 'response to referees' comments document.
- 3) Line numbers in your main document.
- 4) Please read our data sharing policies to ensure that you meet our requirements <https://royalsociety.org/journals/authors/author-guidelines/#data>.

Sincerely,
Dr Sasha Dall
mailto: proceedingsb@royalsociety.org

Associate Editor Board Member
Comments to Author:

The paper has been seriously modified since the first version and unfortunately although the new analysis is more thorough, some of the original interest has been lost (see review 2). Both reviewers make a long list of suggested changes and questions. I agree that these changes are important, but in revising the manuscript, I suggest that it is important not to lose sight of the original version, which included some good/interesting ideas even if questions were raised.

Reviewer(s)' Comments to Author:

Referee: 2

Comments to the Author(s).

I enjoyed reading the revision of this paper, and thank the authors for the extremely hard work they put into improving this paper. The model and the presentation are both much clearer, although I still have some significant questions and suggestions on both. Some of the newly added sections could use a bit more editing for consistency and conciseness, but I didn't mark all of those spots.

As I was reading, I found it odd that the deterministic model and stochastic model are in different units. Why not just do everything in cell numbers? The deterministic model will scale, so the results will be identical, and save readers from making one more translation.

The paper discussed this, but it is hard to get a sense of the diverse trajectories followed by patients either from the medians or the quartiles. Would it be possible to see some examples of actual patient data?

I have two remaining concerns about the model. First, I can't see why the saturating term in the B equation has $k_B + C$ rather than $k_B + B$ in the denominator. The latter produces an upper limit on the killing rate per CAR T cell. With a small value of C, for example, the per capita killing rate increases linearly without bound in B with the existing form.

Second, the $rC(T)$ term is not well motivated. The whole equation still depends only on T, and thus this just gives a different form of competition. I can't see where this form comes from. It just magically appears in the transition from equation (1b) to (2b) in the supplement. What does "immune reconstitution capacity" mean in Table 1? Why is the Gompertz form with lower carrying capacity not sufficient? Page 4 (lines 22-23) seems to be hinting at something along these lines, but need some justification. In addition, it looks from Figure 2C that the results are not very sensitive to the choice of a and b, so perhaps this whole term could be eliminated or simplified.

The paper seems a bit inconsistent about the alternative sources of CD-19. On page 4, it sound like 50% of patients do have detectable normal B cells in circulation. That does not sound negligible to me.

I am concerned that the model results are most sensitive to the values of the carrying capacities. These are problematic parameters because they have little mechanistic interpretation, and I don't see how they could be manipulated in any way by treatment. Although it is perfectly reasonable, it is hardly surprising that increasing the tumor-killing rate of CAR T cells would improve outcomes.

An additional problem with models based on carrying capacity is their failure to separate birth and death rates, such as are discussed on page 7. These almost certainly could not be identified from this sort of data, but interpretations should be more cautious because the negative birth rates that occur when $T > KC$ cannot be treated as death rates.

I found the justification of the Gompertz form for immune system growth a bit confusing. Why would tumor growth laws help us understand immune system growth? I'd shorten this part, and just say that it provided a good fit.

The final paragraphy doesn't make a very strong conclusion. That extinction is absorbing in a stochastic model even when it might be unstable in a deterministic model is well known. I'd want to end with something more like "Effects of CAR T are transient, and we therefore need to maximize their initial impact and knock tumor cells into the stochastic regime as quickly as possible. Our models describe these dynamics, and point toward mechanisms that might enhance the effectiveness of this therapy in eradicating tumors."

Minor points:

Abstract, line 19: I'm not sure if this is some technical phrasing, but "cure events" sounds awkward. Why not just "cures"?

Line 21: "progression occurs much later and is widely distributed in time. We parameterized our model with time series of CAR T cell populations and tumor size and..."

Line 25: I don't think we can quantify "why". I'd go with "how".

Page 2:

Line 4: Unclear what "Relapsed and refractory" mean here.

Line 5: I'd say that CD19 is a marker rather than a protein.

Line 6: "LBCL patients who do not respond to chemotherapy have a median overall survival of under 7 months".

Line 12: What does "would have been" mean here? I don't see how this lines up with the 82% and 54% above.

Line 30: "stochastic due to small population size of tumor cells."

Line 31: "explain" is a bit strong, perhaps "match". Perhaps "reveals" could be "predicts". Finally, the sentence starting on line 32 doesn't add much and I think could be deleted.

Figure 1A: "co-evolution" is a strange term for the interaction between N and C. Isn't it just competition?

Page 7, line 5: "disfavored" is awkward.

Page 7, line 6: This is pretty wordy. Competitive exclusion works, so I'd cut this sentence and rewrite the next sentences .

"This CAR T cell disadvantage predicts their decline and eventual extinction. Other mechanisms, like CAR T cell effector memory cells could maintain persistence."

Page 7: line 26. How about "Stochastic tumor extinction may explain observed treatment success rates". This section should be more clear that a deterministic model cannot drive any state variable to zero, and thus the tumor will necessarily bounce back. This paragraph could end more clearly "brought down to very small cell numbers and subject to stochastic extinction."

Page 8, line 1: I don't see why times would follow a broader distribution based on the previous argument. And lines 11-12 on this page could bear with more discussion.

Figure 3: min, median and max are not defined clearly in the figure caption. In that caption, what is adaptability? I might have missed it, but what is the ALC of the six cells?

Page 10, lines 2-3. I'd delete the new phrase and just end with "However, the tumor growth rate might decrease with tumor burden."

Page 10, line 16: Not sure what "driving event" means.

Page 11, line 10-12: I couldn't follow what is being said here.

Page 11, line 21-22: I'd cut the phrase "with which additional complexity..." And rewrite the next sentence

"Our definition of progression predicts progression events later than those observed in some patients."

Page 11, line 29-32: I found this rather confusing also.

Page 12, line 2: This bit on cytokines seems rather out of place.

Referee: 3

Comments to the Author(s).

In the original manuscript, the model was presented with 3 potential interests: 1/ discriminating possible immune mechanisms responsible for CAR T cells dynamics in vivo, 2/ interesting mathematical properties of the 3-compartments mathematical model and 3/ prediction of treatment optimization with re-injection of CAR-T cells thanks to the model, or predictions of consequences of the model in term of cure or relapse.

In the first round of review, none of these 3 potential interests were addressed convincingly enough and both reviewers raised technical concerns about the mathematical analysis.

Now, after re-submission, the authors solved the technical concerns by re-drawing a new-simpler model with less parameters, that looks convincing enough on the technical level, and is more beautiful/adequate, especially compared to other extremely complicated models in the literature.

However, the authors did remove the parts related to the 'interesting points': 1/ saying their initial claim for one mechanism was controversial so they removed the mechanistic interpretation, and replaced by another controversial interpretation that the carrying capacity is THE reason without explicitly supporting it (see point below); for point 2/ by saying they

disagree that a more deep analytical / mathematical analysis of the model would be beneficial and 3/ by removing the figure related to predicting a new therapy. Also, they discarded the question to assess patient heterogeneity from the data.

So the main statement remaining in this paper from the original version, is 1/ to provide a simple mathematical model to explain the the curves of CAR T cell dynamics versus non-CAR T cells from real patient data when there is B cell depletion before CAR T cell therapy, and (newly added) 2/ to describe stochastically when are the expected times of cure versus relapse although this is based on an 'average patient'. Which is already something adding value to previously published papers (that the authors fairly acknowledge), but lacks a bit the 'meat'.

I want to mention that the work to rewrite the manuscript and keep the track changes is very much appreciated, and the readability was already pretty improved.

I therefore think it would be fair to publish this study but with major substantial modifications.

Major points

1- Assess patient heterogeneity and its effect into the stochastic predictions of cure. I accept that not all data points are available, but there are methods to cope with it, would it be a bootstrap on the variability of each time-point, or the comparison of the few longitudinal available patients. This point is critical, as the prediction relies on stochastic variability. In particular, would any patient have the same chance of relapse, or different parameters inferred from different patients would predict different outcome? Meaning: is stochasticity winning over individualized parameters? This would be a critical point for therapy monitoring.

2- Concluding that the carrying capacity of CAR T cells is different than the one of endogenous T cells would need hypothesis comparison. This is true that the model suggests it is possible to explain the dynamics with two different carrying capacity, but does not exclude that other hypothesis could equally well explain the data (especially as the data has not so many time-points). Please thoroughly discuss other possible hypotheses (as the ones I already raised in the first round of revisions), and replace the strong statement by "In the model, different carrying capacities are needed" or "different carrying capacities are consistent with the observed dynamics". - or find ways to better support this statement.

Minor points:

1- Please show the identifiability (profile likelihood?), and parameter correlation as figure, not only in the text

2- Figure 1D is identical to the original manuscript, please update with the new model

3- Supplementary Figure 1 in the ESM is not visible,

4- Please incorporate better the findings of the ESM into the main manuscript. If not the ESM is fairly unmentioned although it brings valuable insights.

5- Define Gompertzian,

6- The results start at Figure 2C, while Figure 2A is already a result. Please take time to explain all Figure 2 in the beginning of the results (eventually, move this part out of the methods)

Author's Response to Decision Letter for (RSPB-2020-2765.R0)

See Appendix B.

RSPB-2021-0229.R0

Review form: Reviewer 2

Recommendation

Accept with minor revision (please list in comments)

Scientific importance: Is the manuscript an original and important contribution to its field?

Good

General interest: Is the paper of sufficient general interest?

Good

Quality of the paper: Is the overall quality of the paper suitable?

Good

Is the length of the paper justified?

Yes

Should the paper be seen by a specialist statistical reviewer?

No

Do you have any concerns about statistical analyses in this paper? If so, please specify them explicitly in your report.

No

It is a condition of publication that authors make their supporting data, code and materials available - either as supplementary material or hosted in an external repository. Please rate, if applicable, the supporting data on the following criteria.

Is it accessible?

Yes

Is it clear?

Yes

Is it adequate?

Yes

Do you have any ethical concerns with this paper?

No

Comments to the Author

Again, I must commend the authors for the hard work they have put into this revision and the care in responding to a variety of sometimes contradictory comments. I only have a couple of minor points to make about this revision.

Minor points:

Abstract, Line 30: "before a second CAR T injection"...

Page 4, line 5: Something is out of place with the "both can be transformed" clause.

Might be worth starting the Discussion with a bit of context for those readers who skip straight there.

Page 13, line 1. The sentence starting "Our results" is confusing, and maybe isn't even needed.

Page 14, line 9-14. This struck me as a little odd, and either out of place (the last sentence) or not needed.

Review form: Reviewer 3

Recommendation

Accept as is

Scientific importance: Is the manuscript an original and important contribution to its field?

Good

General interest: Is the paper of sufficient general interest?

Excellent

Quality of the paper: Is the overall quality of the paper suitable?

Good

Is the length of the paper justified?

Yes

Should the paper be seen by a specialist statistical reviewer?

No

Do you have any concerns about statistical analyses in this paper? If so, please specify them explicitly in your report.

No

It is a condition of publication that authors make their supporting data, code and materials available - either as supplementary material or hosted in an external repository. Please rate, if applicable, the supporting data on the following criteria.

Is it accessible?

Yes

Is it clear?

N/A

Is it adequate?

Yes

Do you have any ethical concerns with this paper?

No

Comments to the Author

I appreciate the efforts from the authors to improve clarity and take into consideration the points raised at the last revision. I don't want to further delay publication.

Decision letter (RSPB-2021-0229.R0)

26-Feb-2021

Dear Dr Altrock

I am pleased to inform you that your manuscript RSPB-2021-0229 entitled "The roles of T cell competition and stochastic extinction events in CAR T cell therapy" has been accepted for publication in Proceedings B.

The referee(s) have recommended publication, but also suggest some minor revisions to your manuscript. Therefore, I invite you to respond to the referee(s)' comments and revise your manuscript. Because the schedule for publication is very tight, it is a condition of publication that you submit the revised version of your manuscript within 7 days. If you do not think you will be able to meet this date please let us know.

Online supplementary material will also carry the title and description provided during submission, so please ensure these are accurate and informative. Note that the Royal Society will

not edit or typeset supplementary material and it will be hosted as provided. Please ensure that the supplementary material includes the paper details (authors, title, journal name, article DOI). Your article DOI will be 10.1098/rspb.[paper ID in form xxxx.xxxx e.g. 10.1098/rspb.2016.0049].

[http://datadryad.org/submit?journalID=RSPB&manu=\(Document not available\)](http://datadryad.org/submit?journalID=RSPB&manu=(Document%20not%20available)) which will take you to your unique entry in the Dryad repository. If you have already submitted your data to dryad you can make any necessary revisions to your dataset by following the above link. Please see <https://royalsociety.org/journals/ethics-policies/data-sharing-mining/> for more details.

Sincerely,

Dr Sasha Dall

Associate Editor

Comments to Author:

Thanks for taking care of the revisions - the reviewers agree that the paper is ready to go - just a very few, very minor corrections to look at.

Reviewer(s)' Comments to Author:

Referee: 3

Comments to the Author(s).

I appreciate the efforts from the authors to improve clarity and take into consideration the points raised at the last revision. I don't want to further delay publication.

Referee: 2

Comments to the Author(s).

Again, I must commend the authors for the hard work they have put into this revision and the care in responding to a variety of sometimes contradictory comments. I only have a couple of minor points to make about this revision.

Minor points:

Abstract, Line 30: "before a second CAR T injection"...

Page 4, line 5: Something is out of place with the "both can be transformed" clause.

Might be worth starting the Discussion with a bit of context for those readers who skip straight there.

Page 13, line 1. The sentence starting "Our results" is confusing, and maybe isn't even needed.

Page 14, line 9-14. This struck me as a little odd, and either out of place (the last sentence) or not needed.

Author's Response to Decision Letter for (RSPB-2021-0229.R0)

See Appendix C.

Decision letter (RSPB-2021-0229.R1)

02-Mar-2021

Dear Dr Altrock

I am pleased to inform you that your manuscript entitled "The roles of T cell competition and stochastic extinction events in CAR T cell therapy" has been accepted for publication in Proceedings B.

Your article has been estimated as being 9 pages long. Our Production Office will be able to confirm the exact length at proof stage.

Open Access

Paper charges

Sincerely,

Proceedings B

Appendix A

Dr. Philipp M. Altrock
Assistant Member
Department of Integrated Mathematical Oncology
H. Lee Moffitt Cancer Center and Research Institute
Tampa, Florida 33612
philipp.altrock@moffitt.org

Sasha Dall, PhD
Editor
Proceedings of the Royal Society B

March 22, 2021

Re: Rebuttal and resubmission of our manuscript RSPB-2020-1301, new title: " Response to CAR T cell therapy can be explained by T cell dynamics and stochastic extinction events"

Dear Dr. Dall,

Please find attached our heavily revised manuscript, now titled "*Response to CAR T cell therapy can be explained by T cell dynamics and stochastic extinction events*", which we are re-submitting to be considered as a research article in the *Proceedings of the Royal Society B*. This manuscript was kindly reviewed by three anonymous referees, and rejected in its original form, but invited for a resubmission.

In summary, the three referees gave constructive criticisms that mostly focused on shortcomings of presentation and proper definitions, as well as on model choice. All referees remarked that definition and role of the CAR memory cells were not well described. It was also remarked multiple times that the overall presentation was difficult to follow. We believe that we were able to fully address these general points: we have chosen a simpler modeling framework that does not suffer from potential overfitting, and we have thoroughly rewritten the paper and the supplementary material. In one particular comment, referee 2 remarked that "Any model where a tumor or population in general is rapidly reduced and then rebounds can be explored in this way. What is different about CAR T cells from any other mechanism, and what do we learn from the modeling that we didn't know in advance?" We indeed believe that this is a striking difference in comparison to other models of treatments: many models assume, perhaps wrongly, that tumor eradication is a deterministically stable steady state with a finite, non-zero basin of attraction. We show that this assumption is not required in the case of CAR T cell therapy. One could speculate that this pattern is relevant in other mathematical modeling approaches of cancer treatment, and that stochastic extinction events are understudied.

In the revised manuscript, we have rigorously addressed the referee's points by choosing a simpler model, as well as removing speculation about complex mechanisms. We are thus able to provide a more elegant, simpler framework that still captures the essence of biological interactions that we believe are important, and that have not been considered in previous attempts to quantitatively model CAR T cell therapy dynamics. Specifically, we now model a single CAR T cell compartment that interacts with normal T cells, and we still model stochastic extinction in the tumor as a result of tumor killing by these CAR T cells. We believe that this then avoids potential model overfitting: we can explain the normal T and CAR T cell kinetics (10 data points) plus the tumor state at a specific time point (1 additional data point), using 9 parameters, of which 2 are taken from the literature and 7 are fit. We also now perform a parameter sensitivity analysis, to rank these 9 parameters with respect to their impact on the system, which reveals that not only tumor killing by CAR T cells, but also CAR T cell's carrying capacity could be meaningful targets to improve therapy.

Overall, we want to thank you and the referees for the very useful comments and criticism. We firmly believe that we were able to improve our manuscript accordingly, as can be seen in the attached file. The following pages contain the point-by-point reply to the three referees.

Sincerely, on behalf of all authors,

Dr rer. nat. Philipp Altrock

In the following, we give a point-by-point reply to the points provided by the three referees. Each point is in *black italics*, our respective responses highlighted in **green**.

Referee: 1

Comments to the Author(s)
Major Comments:

1. The role of memory cells is unclear. As described, the model pools naïve and memory populations of normal and CAR T cells. Presumably all subsequent references to memory populations are in fact references to the naïve-memory pooled population.

We have removed the partitioning of CAR T cells as our currently available data limits the confidence we can have in the parameters required for a four-compartment model.

2. The authors argue that the growth rate of the CAR T cells should decline after some time τ , and suggest that a mechanistic explanation for this property is attack by normal T cells. However, the change in the growth rate is independent of the population of normal T cells. This may not be relevant to the arguments of the paper if this decline in growth rate is adequately supported empirically.

This is a good point. We have addressed this in two ways to be in line with another critical point brought up by another reviewer. We have removed the explicit time-dependence on the growth rate. In addition, we suggest that the decreased signal of proliferative cytokines caused by the ALC returning to homeostatic value is what accounts for this shift in the changing turnover rate.

3. The model allows for (and finds through fitting) different carrying capacities for normal and CAR cells. How do the authors interpret this biologically?

We have added a paragraph in the discussion section that offers a potential explanation for why this arises. We believe that it is primarily due to the mal-adaption caused by the engineering process to create the CAR T. This can be due to unintended changes at the receptor-binding level that causes it to not be as efficient at utilizing cytokine signaling or nutrients in the environment as compared to its unaltered counterpart.

4. The measurements used for model fitting appear to be: five time points of lymphocyte counts, six time points of CAR T cell measurements, four time points of tumor size estimates. Median values appear to have been used in all cases, which I interpret to mean that fifteen data points are used to fit the data. The number of parameters fit in the model is sixteen. This raises concerns about the identifiability of the model parameters. As acknowledged in Figure 2 B and 2C, multiple parameterizations fit the data, with a range over τ that seems to cover a range of biologically meaningful outcomes. While the scope of the model has the benefit of proposing mechanistic explanations, a reduced-order model may provide stronger support to the claim. If the authors' key claim is that the timing (τ) of decline in CAR T expansion rate predicts outcomes, then the simulations can provide a plausible mechanistic basis and plausible parameter ranges, but a phenomenological model fitting only τ might be sturdier.

This is an excellent point. Based on the available data, we have considered a reduced model that is only three compartments (a single compartment for CAR). The model now contains nine parameters and 11 data points. We had initially used multiple combinations of tumor volume at later days (e.g. 90, 180), but as these measurements were qualitative with a wide range of possible quantitative values, we kept it to only the first clinical visit (day 30). We fit the model based on whether the patient was CR at day 30 (no detectable tumor) or whether they had progressed (20% increase in initial tumor size).

Referee: 2

1) *The presentation itself is extremely difficult to follow.*

We have removed a lot of extraneous detail and have (hopefully) considerably improved the readability of the manuscript.

2) *The model includes non-autonomous terms that for me call into question the importance of the mechanisms.*

This is an excellent point and we thank the reviewer for bringing this to our attention. We have modified the equation to be fully autonomous by arguing that the growth rate can perhaps be better interpreted as the time scale on which either birth or death events occur. In this setting, the need for placing dependence on r_c , we believe, is clearer. As the T cell population returns to its homeostatic value, the cytokine signals that control turnover rate are returning to normal levels. We argue that the dwindling levels of proliferative cytokines (seen by proxy to go down since ALC levels off within a few weeks) shifts the CAR T into a more quiescent state which is what causes the slower decay rate.

3) *I haven't attempted to point out all the places where the presentation is difficult to follow, but think that a clear description of the data available, followed by a far shorter and more direct exposition of the deterministic model would set things up to present the key results and motivate the stochastic model. I found myself wondering why the paper seems so complex when the strength of the approach is precisely the opposite.*

We have cut a massive amount of detail and hope that the revised manuscript is much more concise and we agree that the approach is meant to show the opposite and apologize for the lengthy description.

4) *The non-autonomous terms on page 3 are introduced as being "of note", but with no motivation for not making them part of an autonomous model. If indeed the mechanism is attack by "wildtype" T cells (a term not used before in the paper) then why not include that autonomous mechanism? Models like this are about mechanisms. At the bottom of page 6 there is a key point about timing that I think the paper should be focused around directly, with explicit modeling of the immunogenic mechanism and ideally some alternatives.*

As stated above, we have changed this term to make the model fully autonomous. While we could introduce a predation term by the T cells, this would require at best a second T cell compartment that begins filling once presented with the antigen. However, due to the lack of sufficient amount of clinical data, it would be hard to put much stock into the obtained fits (due to potential overfitting). Note that we have revised the entire model to still reflect normal and CAR T cell competition, but also to avoid potential overfitting due to the now removed second compartment of CAR T cells.

5) *I am not sufficiently familiar with these treatments to say, but I found the emphasis on using the model to test the effects of a second dose rather poorly supported, particularly if we do not understand the mechanism leading to the apparent immunogenicity. The scenarios at the end of page 10 seem kind of a stretch, and I do not think this application is necessary, particularly if removing this would help to reorganize the paper.*

Thank you, this is a good point. A far richer data set would be required to allow for cross-validation and then an actual test on the remaining set for the addition of a second treatment. We have removed this section of the paper.

6) *At the end, the more I thought about this paper, the more unclear it became what the critical predictions are. Any model where a tumor or population in general is rapidly reduced and then rebounds can be explored in this way. What is different about CAR T cells from any other*

mechanism, and what do we learn from the modeling that we didn't know in advance?

This is an interesting question and we have tried to highlight what makes this system and modeling approach a useful alternative to the ones typically analyzed. Often, dynamical systems involving the immune system will have the “tumor-free” state as a potentially stable state subject to some conditions on the parameters. However, when the drug (e.g. CAR or chemotherapy) will ultimately decay over time then what can the mechanism of cure be? We could introduce an Allee effect on the tumor that causes it to go extinct when below a critical size, but we would argue that such an effect introduces a strong condition that we do not need to make. Indeed, a natural criticism of this would be what caused it to go over this threshold in the first place? The one we chose to employ is precisely what makes this a useful modeling approach.

We believe that our type of (hybrid) modeling approach is very useful for future studies in which a majority of the system can be analyzed deterministically, while a small subset can be done stochastically. As many treatments fall under this framework, we believe a broad range of systems would fall under this category.

7) Abstract: I find it confusing when CAR is used without the "T". I'm not sure if this was a mistake or a misunderstanding by me. There are many terms used without being introduced: "immune reconstitution dynamics" and immunogenicity are examples. This problem is even more severe in the Introduction as noted below.

We have changed it so that everywhere in the text, we use CAR T. We have also removed the discussion on immunogenicity as justifying this potential explanation for the change in time scale required too many additional parameters and a second compartment for specialized T cells designed to target the CAR T.

We have removed the reference to reconstitution dynamics from the abstract and explain it in more detail in the subsequent text.

8) "rate-disadvantage": Pretty sure the hyphen is in the wrong spot.

This was removed, thank you.

9) Overall response and complete response are not defined, and I can't see how these numbers are consistent with the 60% at the end of the first paragraph.

We have added more description of what the clinical definition is for OR and CR.

10) Is the mixed effect modeling statistical modeling?

Yes, but we removed that description of Stein *et al.*'s (Ref. 6) modeling, now describing their approach in a little more detail, word limitations permitting.

11) At the end of the page, "homeostatic niche" is not defined, and it is not yet clear what feeds back to CAR T cell differentiation.

We have added more detail on what we mean by homeostatic niche. By homeostatic niche, we actually mean the homeostatic level of T cells. We have clarified this point in the main text and hope it is clearer. Now that we have removed the two compartments that comprise CAR T cells, the differentiation feedback no longer needs to be defined.

12) It would be good to provide more justification for the choice of these particular four state variables.

To introduce a minimal mechanistic modeling approach, we believed that these four (now three)

components were the minimum amount of information needed to recapitulate dynamics and median outcome of patients. The justification for the partitioning of the CAR into two subsets we believed was warranted based on literature review of antigen stimulation on T cells and the fact that the composition of the bag based on naïve/memory vs. effector T cells played a role in clinical outcomes. However, based on overfitting and lack of sufficient data, we have amended the description and now consider CAR T as a single compartment.

13) *The use of a form that I think leads to Gompertz growth is justified to some extent in the supplement. Something about this should definitely be included here.*

Good point! We have added more description of why Gompertzian growth may emerge biologically (as well as numerically) in the main text.

14) *All the material on CCR7 comes completely out of the blue! What does this mean?*

This material was in regards to distinguishing between the two major CAR T compartments (naïve/memory vs. effector) that we introduced in the previous version of the manuscript. In the modified manuscript, the CAR T compartments are now merged into a single compartment and so this problem has been removed.

However, we apologize for not introducing CCR7 appropriately—CCR7 is a marker for T cell differentiation.

15) *Why choose the min functional form in the equation for r_E ?*

We originally proposed this function based on unpublished data that showed a decrease in favorable clinical outcome at higher tumor burdens. A natural place to tie this was at the differentiation level, we could have also tried a Hill-type function or any other sigmoidal. Regardless, with differentiation removed from the new model, this has been removed.

16) *Why is it "unlikely" for $K_m > K_n$? (also, I think the capitalization of the subscripts is inconsistent).*

This is a good question, that we answered in the SI, but perhaps should have relegated a note on in the main text. If the CAR T memory's carrying capacity exceeded the carrying capacity of normal T cells, then linear stability theory predicts that the normal T cells would ultimately go extinct. We believe that this does not occur, hence, in the context of our model, it must be that K_M (now K_C) is smaller than K_N . All subscripts should be capitalized. We added a sentence on this point to the 4th paragraph of the results section.

17) *I don't think papers need to say that things are "novel" -- the results should speak for themselves. And there is a long tradition of immunological modeling that doesn't look fundamentally different from this, which is not at all a bad thing. I did a quick search and found the following for CAR T therapy. They might not be very appropriate, but there is a lot of literature in this area.*

We thank the reviewer for bringing these references to our attention. We have removed the emphasis on novelty, or "first" (removed from abstract), and hope our results speak for themselves as they contribute importantly to the field.

In regards to the papers introduced, the first one (Mostolizadeh *et al.*, 2018) uses a very different approach. They take an optimal control perspective based on some clinical data from over a decade ago, combined with biological intuition and structural requirements of their model. It is not clear which parameters were directly inferred in the paper, and which are consequences for modeling feasibility. Mostolizadeh *et al.* also considered the influence of healthy B cell feedback,

which is an interesting additional component, but since patients in ZUMA-1 did not have healthy B cell return until 90 days (as explained in the main text), we felt it was unnecessary to include additional parameterization to our model. We do not feel comfortable citing and explaining this paper at this time, but are grateful that the referee mentioned it to us.

The second paper (Hardiansyah *et. al.*, 2019) designed a model with 20 (!) parameters with 7 being fitted by the optimization routine. It is conceptually very similar to our original model (before we removed the second CAR T cell compartment), with the exception that they have data for three important cytokines (IL-6, IL-10 and IFN-G). However, to avoid overfitting, Hardiansyah *et. al.* made a few assumptions that we feel are not completely justified. First, they assumed CAR T cell parameters are similar to regular T cells, when in stimulations is not adequately justified. We believe that clinical and *in vitro* (Sahoo *et al.*, 2020) data show evidence to the opposite. Second, CAR T cells are assumed to be poorly adapted compared to regular T cells, which we also find. Further, it is unclear why tumor is eliminated in a deterministic model while one sees the elimination of the CAR T cells. A possible explanation could be that in the disease studied, which is CLL, patients have CAR T that survives longer *in vivo*.

We have added a brief discussion of the second paper to our main text.

18) *The second half of the first paragraph seems out of place, and should be integrated into the flow of the paper.*

Done.

19) *Throughout, I found the use of the term co-evolutionary distracting. What we have is coupled dynamics, with no evolution in the narrow sense.*

We have changed the term co-evolutionary to co-ecological competition.

20) *The parameter estimates in the middle paragraph are not given with any citations.*

We have removed this paragraph. All parameters that were not estimated in this work are given a reference in the table at the end of the main text.

Page 7:

21) *I don't think this is an extinction vortex -- just extinction due to demographic stochasticity without the addition factors that characterize the vortex, such as increased variability or inbreeding.*

This is a good point. We have renamed it to just be extinction due to stochastic fluctuations.

22) *The sentence starting "In particular" seems garbled and is definitely awkward.*

Changed, thank you.

Figure 3:

23) *I was confused by the labels of panels D and E as "Survival" in the title and "Probability of cure" on the vertical axis.*

We have changed it to be clearer. This figure has also been changed due to our changing of the model.

Page 9:

24) *Is there any data to predict the degree of patient variability? I would imagine it could be quite large.*

We currently do not have access to data that would allow us to predict the degree of patient variability, but the reviewer is correct, we would expect certain parameters to be quite large. We have added parameter estimates for the first and third quartiles and added discussion of this to the main text (table 1). Of note, we currently are not able to answer whether patient variability is large or not. However, we have added a discussion of the parameter sensitivities to our analysis (see Fig. 2 C and Results).

Page 11:

25) *The Discussion jumps right into the middle, and should begin with a summary of the key questions and findings, including defining what patient outcomes are being discussed. The problems with terms that have not been introduced or motivated reappears here.*

Good point. We added a summary of key findings and utility of our modeling approach.

Page 12:

26) *Many paragraphs end with generic statements about the value of the modeling approach, or that more modeling work is needed in the future. For example, why integrate cytokines? What does the model miss? As a detail, I thought that IL-7 and IL-15 were less inflammatory and more immunoregulatory.*

The integration of cytokines could help elucidate patient variability and explain why treatment works well in some patients and not others. IL-7 and IL-15 are immunoregulatory, but they promote growth and so could help explain the difference in overall response of patients.

Page 13:

27) *I was a bit confused by the statement that cures "way past 100 days" (which is a bit too casual a term) are driven by some other mechanism. This puts a lot of faith in the model getting it right for the most common mechanisms, which I don't see being justified at this point in our understanding.*

We respectfully disagree that this level of faith is too high for the current state of the model. The fact that the model (the original and revised model) predicts cure to occur before this time is simply due to the fact that CAR T cells are not present in the majority of patients after 90 days. Over 75% of patients have less than 0.5 CAR T cells/ μ L at day 90, which is around the level at infusion. The primary mechanism is tumor killing through CAR T-tumor interactions and this is not an unusual mechanism. A consequence of this mechanism is that the model predicts that a probability of cure falls off dramatically as the CAR T cells decline. Thus, a natural prediction emerges that any patient who is not a complete responder before day 90 but becomes one at a later time point likely benefits from some additional mechanism, such as a vaccine effect in which normal T cells adapt predated on CD19 positive cells. This aspect of CAR T cell therapy cannot be addressed using the current clinical data situation (see also next comment).

28) *The final paragraph loads up a lot of immunological detail, which seems out of place. The paper ends with a rather generic statement that models are useful. What have we learned and what specific questions has the modeling opened up?*

This is a good point and we have removed these details in the revised manuscript. Since we do not have this level of detail in the data, it adds little to the text.

29) *Not sure if this is a typo or if I didn't understand, but the lines on N and M+E seem*

inconsistent ($3.0e9/6 = 5e8$ and $1.8e9/0.36=5e9$).

Apologies, this was a typo. The simulation and results all used the correct value of $1.8e8$ for CAR T and $3.0e9$ for normal T Cells.

Supplementary Material:

1) The first paragraph of the Introduction is almost impossible to follow.

We have rewritten this part and most of the SI for improved clarity.

2) *Why use the median? Are the data time series on individual patients or do only have aggregated data?*

We only have complete data (5 time points) for the quartiles. We have added the parameter estimates for the first and third quartiles as well in Table 1.

3) *As just one example of how confusing the writing is "these counts include CAR T cell density, yet differences in parameters estimates were minimal and no changes in homeostatic ALC were detected" Different in what?*

Agreed, this is ambiguous and has been fixed in the revised SI. The difference was in regards to parameter estimation (M + E + N vs. N). The statement was meant to say that the total amount of CAR being included or not as part of the ALC fit had a minimal impact on parameterization.

4) *The references and powers on page 2 on not properly formatted.*

Fixed. Thank you.

5) *This was very hard to follow: "Patients in none of the categories (CR, SD, PD) after 1000 days would be defined as undetermined, which did not occur in simulations."*

We ran simulations to a maximum of 1000 days, if a patient was not in any of these categories, the simulation classified them as undetermined. We originally thought this may address the question of how we classify a patient that has neither been cured ($B = 0$), nor progressed ($B > 1.2 B_0$) by day 1000. However, since this seems to be unclear, we have removed it in the revision.

6) *How realistic is it to make T cell populations deterministic?*

This is a good question. Both populations are well over 100 million cells from day 0-100. We believe this is sufficiently large that we can neglect fluctuations about the mean. An argument could be made for letting CAR T become stochastic at a much later time, but since data is only available for the first six months, switching to a stochastic framework is not necessary and even incorporating it for later time points should not be important in terms of tying it to clinical outcomes since CAR T go extinct in this model in the deterministic limit.

7) *Page 4: Variance and mean have different units, so I don't see how the variance could be 5-15% of the mean.*

Apologies. This was meant to be standard deviation and has been fixed in the revised version by removing the statement and describing the method in more detail in the ESM.

8) *Page 4: "We can see by plugging in a few choices for ? that the impact of changing the threshold is minimal". This is kind of hard to evaluate. This section is kind of wordy also. If the point is that with an exponentially growing tumor different thresholds occur at similar times, say that. And is that level of growth realistic? The one-day difference cited here seems rather*

extreme.

Tumor doubling times for LBCL range widely from 1-70 days with the mean tumor growth rate around 10-14 days. The section was meant to minimize the concern over choosing a progression threshold different from that established in the clinic. We have revised the text and shortened it for brevity.

9) I'm not convinced that section 3 is needed. If we saw an explicit fit to the data that worked well for some aspects of the data and not for others, that would help understand the system, but this seems like an exercise for the authors.

This is a good point, we have removed this section.

10) Section 4 opens with an unmotivated statement about non-autonomous behavior. As I say in my main comments, such an assumption begs the question, because nothing in biology is a function of time, but is a function of other state variables.

This is an excellent point. We have modified the function to be autonomous, while still containing the same behavior that recapitulates the long-term dynamics of the CAR T. We have updated this part of the SI to mirror this change.

11) Page 5: "Third, the CAR T cell products volume (population size) and composition should be integrated, leading to a CAR T cell population that is sub-divided into at least two populations, for example (central) memory and effector CAR T cells." I cannot understand the logical link.

We agree, this is unclear as written. Additionally, as we now merged the two compartments into one, this statement is even less meaningful. It has been removed. The rationale was based on the fact that certain phenotypic breakdown of CAR T for patients led to more favorable clinical outcomes and this implies the utility of considering the CAR T partitioned into (at least) two types: those that created the tumor-killing cells, and the tumor-killing cells.

12) Page 6: As I also mention in the main section, the terminology of co-evolution is very confusing to people who work in ecology, and I cannot see what is game theoretic about a model of population dynamics.

This is a fair point. Originally in the model development stage, we had considered the interaction at the individual level and arrived at our model through a mean-field limit. In this context, the interaction rates that gave rise to the carrying capacity was due to individual interactions, fitness can be tied to the difference in carrying capacity. Since this has been unclear, we have removed this term and replaced it with other descriptions in the ESM.

13) Supp Fig 1: Heterogeneity among patients can cause the appearance of multiple decay rates. Is it possible to analyze data in this way, or, if not, compare models that assume a changing rate with those that assume heterogeneity of rates?

This is an excellent point. Unfortunately, we do not have the granularity of data to be able to determine if it is the heterogeneity among patients that causes this behavior. It is entirely possible that heterogeneity of rates can recapitulate the expected clinical outcomes, however it clearly would be unable to recapitulate the median patients CAR T trajectory.

14) Page 7: This section is quite wordy and hard to follow. I don't see the system as being that complicated, which is a virtue.

Yes, this is true. We have removed a large portion of this section and highlighted the system's simplicity, which we believe is a strength.

15) Page 8: *It would be good to see some indication of the preliminary analysis arguing that beta is small and r is large. I'm not sure what large means for a parameter with units.*

It is true that “large” is a misleading term. The preliminary analysis that indicated this behavior was that as we increased the size of the bounding box for the parameters, r was always at the largest possible value and beta was as small as possible. Beta is dimensionless and we could potentially try to create some plot that shows how beta and r trend as we increase the bounding box, but we do not believe this adds a lot of value to the text. Indeed, the size of the bounding box for the parameters is usually based on some biological intuition. Once we observed that the optimizer was returning values $\beta \rightarrow 0$ and values for r were in excess of 100 day^{-1} , we replaced the model with the limiting case it was converging to, which is Gompertzian growth.

An alternative explanation is simply that Gompertz yields a better fit because the optimizer is converging to parameters for generalized logistic that lead to Gompertz and so we can afford to lose a parameter.

16) Page 9: *I can't understand why there is no coupling of the memory compartments to anything. For example, there is no loss term when memory cell become effector cells. And why would K_N and K_M be different?*

The memory compartment was coupled to the normal T cell compartment and it is this coupling that leads to CAR T cell removal overall. There is no loss term when memory cells become effector cells because they do not become effector cells, division is asymmetric. Regardless, the model now no longer distinguishes between memory and effector CAR T cells, we have reduced the model complexity.

As for why K_N and K_M (now K_C) are different; we have already discussed this previously that this is a mathematical requirement for CAR T loss over time, but can also be explained biologically – it can be attributed to the CAR T product being mal-adapted through the engineering process. It would be a special case to assume that $K_N = K_C$.

17) Page 10: *I don't think the conditions for a peak need to be presented.*

Removed.

18) Page 10: *"some of these patients" is rather vague.*

This statement has been removed. Figure 1 mentions that the data was obtained from Moffitt patients treated with CAR T cell therapy.

19) Page 10: *I can't see how assuming a fixed fraction of CAR T cells is consistent with the model.*

There is nothing inconsistent with the model to suppose that the memory fraction of CAR is the same proportion throughout. It is a strong assumption though, but the alternative would have forced us to fit the entire model simultaneously, rather than fit the memory CAR T and normal T separately (which was uncoupled) which would exacerbate the overfitting. Regardless, the model has now combined the CAR T into a single compartment and so this point is moot.

20) Page 11: *Why would CAR T cells numbers be normally distributed? Is this a model of stochastic error, process error or measurement error?*

This assumption was made to get an estimate of the variance to be used in weighted least squares regression. It is likely that the high values of CAR T at early time points have a larger variance and so Gauss-Markov theorem tells us that using variance in weighted least squares leads us to the best estimator. However, on revision we have instead used least squares

normalized by each compartment's maximum value and then adjusted the weights with hyperparameters (described in ESM section 4).

21) Page 11: *In my experience, these models can be sensitive to estimates of carrying capacity. Are results robust if K_N is not 500?*

This is an excellent point, in the revised manuscript we conducted sensitivity analysis on all parameters (including those taken from the literature) and found that the carrying capacities were the most sensitive parameters.

We have added a panel in figure 2 of the main text where we show the sensitivity of the CAR T carrying capacity on the model. As expected it is highly sensitive. With the fitted parameters and varying only K_c , we observe a sharp transition around the fitted value in the probability of cure.

22) Supp Fig 2: *It is hard to justify, or check, introduction of a whole new parameter to fit a single day point.*

Apologies for the confusion, but this was not to fit just a single time point, day 180 was not shown (there are five total time points for CAR T). This has been removed.

23) Supp Fig 3: *Would it be possible to show how parameter estimates covary?*

We have added sensitivity and identifiability analysis to the revised manuscript. The latter includes a correlation matrix of the parameters of which we find that the tumor compartment parameters are the least identifiable (γ_B correlates strongly with r_B – tumor growth rate and k_B – a new parameter which is the tumor size needed for half-maximal killing rate).

Referee: 3

1) *In order to claim that 'immunogenicity' is the reason for CAR T cell decay, they should provide alternative hypotheses and show that immunogenicity is most consistent with the dataset. Although immunogenicity is a possible mechanism, the authors didn't support adequately this hypothesis from the literature, therefore I am not 100% convinced that this is the reason.*

Alternative hypotheses:

- *Activated T cells produce IL2 which is a positive feedback for other T cells, so they could actually support each-other, and the limiting factor could be that it is hard to access the tumor (especially if the collagen network is impacted in the zones of the tumor). CAR T cell decay could be that they are exhausted or do not manage to access the tumor cells.*
- *It is not clear how the inject CAR T cell memory pool can sustain the production of effector T cells. Do they have a maximum number of divisions? Do all memory cells become effector with a short life or all becoming exhausted? These hypotheses could explain a synchronized shut down of CAR T cell numbers without needing an 'immunogenicity' hypothesis.*
- *Therefore, the authors should decide whether they want to keep the claim that immunogenicity is the reason (and support it), or remove this claim.*
Note: CAR T cells are supposed to have already high affinity... has anybody shown that late T cells have actually higher affinity than the CAR T cells?

This is a good point by the reviewer. We have removed this as an explanation in the revision and instead introduce a similar, but autonomous feedback on the rate r_c . The rationale for placing this additional feedback (in addition to feedback at the carrying capacity level) is that r_c in effect is the time scale at which birth/death events occur, and as the normal T cells approach their homeostatic level, we would expect the cytokine signaling to return to normal levels and this should lower the turnover rate which is tied to r_c .

It is true that it is difficult to rule out other potential mechanisms and this is the primary reason that we have removed it as the explanation.

2) I have a major concern regarding the fitting of the experimental dataset. Not only the 'average patient' is taken, but the number of model parameter is bigger than the number of data-points, which suggests overfitting. Indeed, Figure 2C, the time of 'immunogenicity switch' shows a bimodal distribution while this is the same fitting repeated multiple times.

- Could the authors provide a more in-depth description of the parameter uncertainty [for instance they do not show the distribution of other fitted parameters]. Are some parameters actually identifiable?
- How does the bimodal distribution impacts on the predictions of relapse times? I want to make sure that the large predictions are not just a consequence of highly different fitting parameters at each fitting.
- I could not understand whether the authors keep a population of parameter sets from the fitting to make predictions, or whether they just take the parameter set shown in Table 1. In the latter case, are there other similarly optimal parameter sets? Do they predict the same?

This is an excellent point. Our original model was meant to be proof-of-concept to be eventually verified by more granular data, but overfitting brings a lot of the conclusions we made into question. Therefore, we propose the simpler model which has less parameters and is no longer susceptible to overfitting, while still retaining the overall shape of the CAR T dynamics.

We have added details on sensitivity and identifiability analysis that shows that all parameters are sensitive with the carrying capacities being the most sensitive. All but three parameters (r_B , g_B , k_2) are identifiable at the 0.9 correlation level. These are all tied to the tumor compartment and it is not surprising that these are highly correlated. This is because we only have qualitative data at certain time points for patients (and at most we have 1-2 time points for tumor).

The new model does not have any noticeable bimodality. We only used the parameter set in table 1 with which we perturbed by a standard deviation that ranged from 5-15% of the mean. The global optimizer, as mentioned in the SI, contains additional parameters that we use to assign more or less weight to particular sets of the data. We were most concerned about recapitulating CAR T dynamics and least concerned about getting exact quantitative fits for the tumor burden as time points past day 0 were qualitative. Changing the weights in the loss function will in general change the landscape and we could expect to obtain different optimality sets.

3) The predictions on the stochasticity of relapse times is based on gaussian distribution around the optimal parameter set (fitted only to the 'median patient'), which doesn't seem to be accounting for patient heterogeneity. [although the detail how they did is well explained in the ESM]. Why not perform a bootstrap on the full dataset at least, to estimate the population variation?

At the time, we did not have the full data set of all 101 patients from the ZUMA-1 trial. At the time of revision, we have received the data set of about 1/5 of these patients, many of which only have 2-3 time points of their CAR T levels. Bootstrapping usually requires the sample to be indicative of the entire population, and with 50% of the 22 patients having 2-3 time points, it subjects these patients to overfitting. A potential workaround would be to fix some of the less sensitive parameters to be at their mean value and change only the most sensitive parameters (in this case K_C , K_N). However, the main focus of this work was on a simple mechanistic model that can recapitulate the dynamics. This type of analysis would be more suited for a complete set with which we can adequately create a train, validate and test partition.

4) The authors predict a large variation in the time to relapse, which is pretty interesting. Could the authors predict the best time-points to measure relapse (like design a schedule that doctors would need to follow). I suggest to look at this work, <https://doi.org/10.1371/journal.ppat.1005535>, that predicts the best time-points to detect HIV relapse.

This is a great idea that we would like to implement in the future after we have laid sufficient groundwork for the model. The paper referenced by the reviewer appears to be a sequel to one in which a mechanistic HIV model was presented, the aforementioned paper describes how they could use this model to better predict the best time-points to detect HIV relapse.

5. I am still not convinced that immunogenicity is the reason for CAR T cell decay, but I think this model can still be useful [and that other mechanism could actually maybe lead to similar equations]. Could the authors predict an experimental setting that would actually answer whether immunogenicity is the reason?

We agree that claiming that immunogenicity was the likely mechanism was a stretch. In the revision, we consider an alternative that is due primarily to the change in cytokine levels as the normal T cell compartment returns to its pre-lymphodepleted levels. This changes the turnover rate of T cells and we claim it is this mechanism that slows the removal rate of CAR T cells so that they are still present months after infusion. This new model does not suffer from over fitting and also replaces the non-autonomous term with one dependent only on the biological values (e.g. on ALC vs. a temporal switch).

6. The model used for the immune interactions is partially analyzed in the ESM but are not linked from the main manuscript, and still I think more analyses of the model property could be beneficial. For instance, do the mutual inhibition interaction motifs confer hysteresis and tri-stability? A more in-depth analysis of the dynamics and the mechanistic consequences of the model design would give a better understanding of the properties of this model.

We respectfully disagree that the model needs further analytical treatment. Given the original model with $K_M < K_N$, CAR T cells are always eliminated from the system in the long run and so you are left with eventually immune reconstitution and tumor growth (unless you see cure via stochastic fluctuations, hence our switch to a hybrid model). The new model which combined the CAR T compartments into a single compartment is no different and does not contain this type of behavior.

In fact, it is this type of simplicity that we believe makes the model more appealing. Despite the deterministic limit predicting progression, the individual-based model can predict cure and depending on parameter choices, can have a majority of patients be cured. The separation of scales for population sizes warrants combining the ODE for large populations and the Gillespie algorithm for smaller population sizes.

7a) The proposition to re-inject CAR T cell is interesting. But since the authors keep the same immunogenicity of normal T cells, the predictions for Figure 4 are pretty straightforward and it is not very helpful to reinject CAR T cells after 20 days. Is there experimental back up to state that reinjection of CAR T cells has no effect on the PFS? It is hard to believe this statement. Now the authors propose to add 2 more steps: reducing the immunogenicity and then injecting CAR T cells again. This sounds pretty heavy for the patient. Could the authors discuss or use the model to propose a different (one step only) therapy? For instance, they didn't talk about exhaustion or check-point. Could they speculate whether checkpoint inhibition or drugs modulating the metabolism of T cells could differentially help CAR T cells versus normal T cells?

There is evidence (Turtle *et al.* 2016) that supports that subsequent infusions of CAR T have minimal improvement due to reduction in peak CAR T. This had been tied to the murine sequence used in the CAR T product. A checkpoint inhibitor would be an interesting alternative that could be given in tandem with the treatment, but without data we could only speculate on

whether this would have an impact, and if indeed it did so, the outcome would be the same as removing the immunogenicity term that we had originally proposed.

Regardless, we have decided to remove this discussion and that of figure 4 because we agree that it is both straightforward and not easily validated. The revised manuscript is to be mostly considered as the presentation of a proof-of-concept for a simple model that despite its lack of a stable tumor-free steady state, its individual-based counterpart contains enough detail to have the model recapitulate the CAR T dynamics and also account for some of the behavior observed with different tumor burden sizes.

7b) Could the authors show whether, for $r_2 = 37.6$ days, the long-term FPS improvement is due to the first prolonged wave of CAR T cells, or whether the second one is actually the reason for it?

The second treatment is no longer discussed, but one way to determine this would be to look at the distribution of cure times between the double- and single-treated patients and see if there is a difference (e.g. via K-S test).

8. In the discussion: «We identified 5 processes as potential drivers of patient outcomes». The authors just mention the mechanisms in the model, but actually do not investigate the effect of these mechanisms on the outcome. Either investigate or do not say they are identified as drivers...

We have removed this statement.

9) There are interesting analyses in the supplementary file, but they are poorly related from the main manuscript.

We have attempted to improve the references between the two documents throughout. Every section of the ESM is referred to.

10) Please provide line numbers in the next manuscript!

Done.

11) Too strong statement in the abstract: The model demonstrate that CAR expansion is shaped by immune reconstitution. You do not prove that (see point 1)

Changed to "...demonstrate that CAR T expansion *can be* explained by immune reconstitution dynamics after lymphodepletion and competition among T cells."

12) Can you explain why only N (normal T cells) and M (CAR memory) share the carrying capacity, but not the effector pool?

This was done for both a biological and modeling choice. First, it is unlikely that during a mounted immune defense the effector cells are subject to any real type of carrying capacity restriction, more likely it is its death rate and subsequent removal of antigen (leading to reduction of new effector cells) that leads to effector cells being removed. Second, by using the first point as a justification, this allows us to uncouple the M and N compartment to be more confident in the obtained fits.

Regardless, the new model considers only a single CAR T compartment C and as such, we believe this is no longer a point of discussion.

13) Figure 2, why not plot the curves for both M(t) and E(t)?

We have replaced M & E with just one compartment C. This is plotted in the new figure.

14) *Table 1: Please give the standard deviation of the fitted parameters.*

We included the ranges based on Q1, Q2 (median), Q3 available. To include standard deviation would require us making the additional assumption that the parameter distributions were normal, which we feel is unnecessary. See Table 1 for the included ranges in the revision.

15) *The authors state that normal T cells replace CAR T cells because they have higher affinity, fine, but then why would the CAR T cells be more efficient at killing the tumor than higher affinity normal T cells? For instance, page 13 they say «Further, detection of normal CD19+ B cells long after CAR T is likely evidence that functional CAR T cells no longer persist in the host»*

This is a source of confusion. We did not mean a higher affinity for the antigen, we mean: better at surviving in the patient long-term. In our modeling framework, this manifests itself through $K_N > K_C$ (or K_M in the old version).

In regards to the second statement, we believe that the fact that normal CD19+ B cells are detected typically past six months post treatment is evidence that functional CAR T no longer persists, this is built into the model as CAR T elimination is a long-term state.

16) *The authors should discuss to which extent their study is proper to LBCL and whether it actually applies to other CAR T cell therapies. For instance, they start after lymphodepletion.*

We believe that the model framework is suitable for any type of CAR T therapy in which lymphodepletion is necessary for expansion. Increasing model complexity is easily achieved through additional compartments that are disease-specific, but for which the appropriate data has to exist.

We have added a brief discussion along these lines to the discussion section, where we discuss recent QSP modeling of CAR T cell therapy in ALL.

—

Again, we thank all three referees for their insightful comments and criticism, which greatly improved the manuscript.

Appendix B

Associate Editor: Comments to Author

The paper has been seriously modified since the first version and unfortunately although the new analysis is more thorough, some of the original interest has been lost (see review 2). Both reviewers make a long list of suggested changes and questions. I agree that these changes are important, but in revising the manuscript, I suggest that it is important not to lose sight of the original version, which included some good/interesting ideas even if questions were raised.

We thank the Editorial Board Member for their positive assessment and have done our best to accommodate their suggestion while further improving the manuscript along the constructive comments raised by the two referees.

Reviewer(s)' Comments to Author:

Referee: 2

Comments to the Author(s).

I enjoyed reading the revision of this paper and thank the authors for the extremely hard work they put into improving this paper. The model and the presentation are both much clearer, although I still have some significant questions and suggestions on both. Some of the newly added sections could use a bit more editing for consistency and conciseness, but I didn't mark all of those spots.

We thank the referee for their positive feedback on the revision of our manuscript and we will address all additional concerns.

As I was reading, I found it odd that the deterministic model and stochastic model are in different units. Why not just do everything in cell numbers? The deterministic model will scale, so the results will be identical, and save readers from making one more translation.

The simulations were all conducted using cell numbers, but the optimization routine used the units from the trial data (cells/ μL). We have modified the table in the main text to contain only parameters using cell numbers.

The paper discussed this, but it is hard to get a sense of the diverse trajectories followed by patients either from the medians or the quartiles. Would it be possible to see some examples of actual patient data?

We now provide examples of CAR T cell trajectories for individual patients in Figure 3A.

I have two remaining concerns about the model. First, I can't see why the saturating term in the B equation has k_B+C rather than k_B+B in the denominator. The latter produces an upper limit on the killing rate per CAR T cell. With a small value of C, for example, the per capita killing rate increases linearly without bound in B with the existing form.

We apologize for not being clearer. We justify the saturation of killing rate with $k_B + C$ by the existence of an upper limit on the amount of CAR T cells that can engage with a particular tumor cell. For example, if 5 CAR T cells can surround a tumor cell, then 50 CAR T cells would not improve the killing rate. Therefore, we used the saturation form described in the main text.

However, a similar argument could imply an upper limit on killing based on larger tumor mass. Therefore, we could have used one of the following functional forms: $\gamma_B B \frac{C}{1+\phi B+\theta C+\sigma BC}$, $\gamma_B \frac{C}{k_C+C} \frac{B}{k_B+B}$. However, given the limited amount of data points, especially for the tumor compartment, it is hard to justify the identifiability of these additional parameters. This is why we ultimately made the choice to keep the factor including CAR T cells only, but we acknowledge that other more complicated functional forms must be examined as more data becomes available to justify their use.

We added a paragraph about the tumor-killing rate and other variables involved in the system in the discussion.

Second, the $rC(T)$ term is not well motivated. The whole equation still depends only on T , and thus this just gives a different form of competition. I can't see where this form comes from. It just magically appears in the transition from equation (1b) to (2b) in the supplement. What does "immune reconstitution capacity" mean in Table 1?

The choice of introducing functional dependence on $rC \rightarrow rC(T,B)$ where the total number of T cells, $T = N + C$, is used to explain the change in CAR T cell decay observed in the data. We can see that there is a period of rapid turnover (days 0-14) followed by a slow decay (days 28-180). One possible explanation is that the CAR T's turnover rate is sensitive to the amount of antigen and the number of proliferative cytokines in the system. As the T cell niche is refilled by both normal and CAR, the number of proliferative cytokines decreases and this is what causes the turnover rate to diminish.

An argument could be made to have $rN \rightarrow rN(T)$ as well, but to avoid overfitting, we chose to place this effect only in the C compartment as it is very likely that the impact on the N compartment is small (very little happens to N after day 28).

Finally, we apologize: the functional form shouldn't have magically appeared, we have changed $rC \rightarrow rC(T,B)$ in equation (1b) in the supplement. We believe the referee is referring to parameter b : Immune reconstitution factor in rC . We tried to find an informative, but short description of the parameter. The nomenclature came from noting that when $T = N + C$ is small, $rC(T) \sim \rho + b$ and so we felt that b could be best viewed as a factor which increases the growth rate – dependent on the current size of immune reconstitution.

We now attempt to better justify the functional form and how it arises in the ESM.

Why is the Gompertz form with lower carrying capacity not sufficient? Page 4 (lines 22-23) seems to be hinting at something along these lines, but need some justification. In addition, it looks from Figure 2C that the results are not very sensitive to the choice of a and b , so perhaps this whole term could be eliminated or simplified.

A lower carrying capacity of CAR T cells is not sufficient to account for the slow decay of CAR T over time after CAR T peak, which is why we introduced the additional terms. In regards to the parameters a, b , we would like to stress that the sensitivities are relative sensitivities; none met the commonly adapted threshold to be removed (see e.g. Olufsen, M. S. & Ottesen, J. Math. Biol. 2013). The fact that they are on the lower end of ranked relative sensitivities does not imply that the model is not sensitive to these parameters.

We added a statement on page 7 to clarify that low relative sensitivity does not imply that the model is insensitive to the parameter.

The paper seems a bit inconsistent about the alternative sources of CD-19. On page 4, it sounds like 50% of patients do have detectable normal B cells in circulation. That does not sound negligible to me.

We apologize for the confusion due to our unclear wording. We believe that both clinically and biologically, that CD19+ normal B cells do not play a role during the first 90 days of CAR T treatment. What we meant to say is that 50% of patients have no detectable normal B cells in the peripheral blood at the time lymphodepletion is initiated, while the entire cohort has effectively zero normal B cells following the lymphodepleting chemotherapy and prior/concurrent to CAR T cell infusion. Furthermore, the reconstitution of normal B cells is extremely slow, such that only 50% have recovered any normal B cells by 1 year after the therapy. While it is possible that normal B cells activate CD19 directed CAR T cells, at this scale we feel confident that the dynamics of CAR T in the context of lymphoma are minimally, and unmeasurably, impacted by normal B cells.

We have changed the wording in the main text accordingly and are grateful for the referee's attention to detail here.

I am concerned that the model results are most sensitive to the values of the carrying capacities. These are problematic parameters because they have little mechanistic interpretation, and I don't see how they could be manipulated in any way by treatment. Although it is perfectly reasonable, it is hardly surprising that increasing the tumor-killing rate of CAR T cells would improve outcomes.

We thank the referee for bringing this up. We respectfully disagree that the carrying capacities have little mechanistic interpretation and understand how our previously inadequate explanation could have led to this conclusion. CAR T cells are created from a polyclonal batch of T cells at various differentiation states, which then undergo artificial stimulation and expansion ex vivo during manufacturing. Therefore, we consider CAR T cells as mal-adapted, at least in comparison to native T cells without gene modification, and this maladaptation can be manifested in the carrying capacity. Making a CAR T product with fewer cell divisions, improving the "stem-ness", or increasing the metabolic capacity of the CAR T cells should lead to a higher carrying capacity. This would lead to a higher peak and a higher total volume of CAR T in the body over the length of treatment.

This description is now incorporated into our manuscript to clarify the mechanistic interpretation of a CAR T cell carrying capacity (page 14).

An additional problem with models based on carrying capacity is their failure to separate birth and death rates, such as are discussed on page 7. These almost certainly could not be identified from this sort of data, but interpretations should be more cautious because the negative birth rates that occur when $T > KC$ cannot be treated as death rates.

This is an excellent point and one that we have spent a good deal of time trying to wrestle with the best interpretation. We believe that the $rC(T)$ term serves to set the time scale on which birth and competition events occur. Whether the term is positive or negative should be interpreted as the net effect of birth and competition. If $dC/dt < 0$, then competitive effects are outcompeting the birth rate, and if $dC/dt > 0$, the birth events are outcompeting the competition effects. The rate $rC(T)$ determines the frequency with which these events occur.

As we have mentioned in previous responses above, this total lymphocyte dependence on rC is a possible explanation for the slow decay, the proliferative signals have shrunken substantially as the T-cell niche has been refilled.

I found the justification of the Gompertz form for immune system growth a bit confusing. Why would tumor growth laws help us understand immune system growth? I'd shorten this part, and just say that it provided a good fit.

We have relegated this to the discussion as we believe this still describes nicely a possible explanation for the emergence of Gompertz even if the paper's developed theory was used to explain tumor growth. The occupation of multiple microstates, e.g. occupying a spatial or functional niche, and a limit of accepting states that have been filled, provides a nice parallel to the selective procedure that the T cells go through during development in the thymus (Klein et. al., Nat. Rev. Immunology 2014?).

The final paragraph doesn't make a very strong conclusion. That extinction is absorbing in a stochastic model even when it might be unstable in a deterministic model is well known. I'd want to end with something more like "Effects of CAR T are transient, and we therefore need to maximize their initial impact and knock tumor cells into the stochastic regime as quickly as possible. Our models describe these dynamics, and point toward mechanisms that might enhance the effectiveness of this therapy in eradicating tumors."

We thank the referee for this observation, have shortened the existing final paragraph and now address the suggested conclusion (with which we agree!) as the final paragraph.

Minor points:

Abstract, line 19: I'm not sure if this is some technical phrasing, but "cure events" sounds awkward. Why not just "cures"?

Changed.

Line 21: "progression occurs much later and is widely distributed in time. We parameterized out model with time series of CAR T cell populations an tumor size and..."

Changed.

Line 25: I don't think we can quantify "why". I'd go with "how".

Changed.

Page 2:

Line 4: Unclear what "Relapsed and refractory" mean here.

This was left over from prior revisions and have removed it.

Line 5: I'd say that CD19 is a marker rather than a protein.

Changed.

Line 6: "LBCL patients who do not respond to chemotherapy have a median overall survival of under 7 months".

Changed.

Line 12: What does "would have been" mean here? I don't see how this lines up with the 82% and 54% above.

Changed "would have been" -> "are".

Line 30: "stochastic due to small population size of tumor cells."

Changed.

Line 31: "explain" is a bit strong, perhaps "match". Perhaps "reveals" could be "predicts". Finally, the sentence starting on line 32 doesn't add much and I think could be deleted.

Deleted.

Figure 1A: "co-evolution" is a strange term for the interaction between N and C. Isn't it just competition?

Yes, agreed. This is better defined as competition.

Page 7, line 5: "disfavored" is awkward.

Changed this to "outcompeted by".

Page 7, line 6: This is pretty wordy. Competitive exclusion works, so I'd cut this sentence and rewrite the next sentences. "This CAR T cell disadvantage predicts their decline and eventual extinction. Other mechanisms, like CAR T cell effector memory cells could maintain persistence."

Changed.

Page 7: line 26. How about "Stochastic tumor extinction may explain observed treatment success rates".

Changed.

Page 7: line 27. This section should be clearer that a deterministic model cannot drive any state variable to zero, and thus the tumor will necessarily bounce back. This paragraph could end more clearly "brought down to very small cell numbers and subject to stochastic extinction."

It is not necessarily true that deterministic models cannot drive any state variable. Certainly $y' = -k*y$ will go to zero (though in infinite time). However, it can even be done in finite time (for example $y' = -k*\sqrt{y}$). In our deterministic model, the tumor only returns because the CAR T cells have been driven close to zero and there is no Allee effect or some other mechanism that deterministically forces small tumors to go extinct. We have changed the end of the paragraph.

Page 8, line 1: I don't see why times would follow a broader distribution based on the previous argument. And lines 11-12 on this page could bear with more discussion.

True, this is merely the observation we make. We changed the wording in the revision.

Figure 3: min, median and max are not defined clearly in the figure caption. In that caption, what is adaptability? I might have missed it, but what is the ALC of the six cells?

Changed to "Initial reduction of normal T cells". Removed the ambiguities, thank you.

Page 10, lines 2-3. I'd delete the new phrase and just end with "However, the tumor growth rate might decrease with tumor burden."

Changed.

Page 10, line 16: Not sure what "driving event" means.

Removed this and replaced with "It is unclear whether B cells themselves could be responsible for continued CAR T persistence."

Page 11, line 10-12: I couldn't follow what is being said here.

We have rewritten this paragraph for clarity.

Page 11, line 21-22: I'd cut the phrase "with which additional complexity..." And rewrite the next sentence "Our definition of progression predicts progression events later than those observed in some patients."

Changed.

Page 11, line 29-32: I found this rather confusing also.

We have changed this for clarification, thank you.

Page 12, line 2: This bit on cytokines seems rather out of place.

We believe understanding cytokines is important for future studies and have explained additional reasons why we think cytokines will be crucial in understanding the dynamics.

Referee: 3

Comments to the Author(s).

In the original manuscript, the model was presented with 3 potential interests: 1/ discriminating possible immune mechanisms responsible for CAR T cells dynamics in vivo, 2/ interesting mathematical properties of the 3-compartments mathematical model and 3/ prediction of treatment optimization with re-injection of CAR-T cells thanks to the model, or predictions of consequences of the model in term of cure or relapse.

We thank the referee for their detailed assessment of our paper's goals, but respectfully object to some of these points. We (i) originally focused on an important, yet overall unresolved aspect of CAR T cell population heterogeneity (memory and effector cells), and (ii) tried to explain the slow decay of CAR T after day 28 via what we falsely called immunogenicity, which we naïvely implemented using a time-explicit CAR T cell expansion rate. This was done in a 4-compartment model (normal, CAR memory, CAR effector, Tumor). Now we present a 3-compartment model. In the subsequent version, and to avoid overfitting as was highlighted by the reviews, we combined the CAR T compartments into a single compartment. However, we believe that we had always been interested in the mathematical properties of our deterministic compartment models, as discussed in the ESM. Thus, in accordance with an important point below, we again have worked to better integrate the ESM results with the main text.

To the final point: we revisited the idea of a second infusion, with or without second lymphodepletion, now using our revised modeling framework (see ESM, Section 6, and Discussion). We can show that a second infusion *without* lymphodepletion would have very limited success. Second lymphodepletion is necessary to elevate a patient from a progressor (with partial response) to a complete responder. Also note that current literature suggests that reinfusion may not yield sufficient improvements due to host immunogenicity against the CAR T cells (Turtle et. al. JCI, 2016), however the assays to detect this are insufficient, and we lack appropriate patient samples to identify whether this mechanism occurs in our patients. Overall, reinfusion and lymphodepletion should be subjects of further investigation, in particular in the context of more sophisticated data and models that consider mechanisms of treatment evasion and resistance evolution, and possibly lower-dose second lymphodepletion, which goes beyond the feasible scope of this manuscript.

In the first round of review, none of these 3 potential interests were addressed convincingly enough and both referees raised technical concerns about the mathematical analysis.

The main criticisms in the first round of review centered on **overfitting**, rather than on lack of mathematical analysis. We believe this was a direct result of using a 4-compartment model. In the second version, in response to the prior reviews, we simplified the model to avoid overfitting and added significant work outlining parameter identifiability and sensitivity analysis, which were absent from our first submission.

Now, after re-submission, the authors solved the technical concerns by re-drawing a new-simpler model with less parameters, that looks convincing enough on the technical level, and is more beautiful/adequate, especially compared to other extremely complicated models in the literature.

We thank the referee for this positive assessment of our simpler model.

However, the authors did remove the parts related to the 'interesting points': 1/ saying their initial claim for one mechanism was controversial so they removed the mechanistic interpretation, and replaced by another controversial interpretation that the carrying capacity is THE reason without explicitly supporting it (see point below);

Our work shows that a mechanism for CAR T cell expansion **and** decay, is the differences in carrying capacities between normal and CAR T cells. The observed normal T cell dynamics suggest that their

process of returning to normal values can be described by growth toward an intrinsic carrying capacity. We now make this point clearer in the methods section of the main text.

In regards to 1/, we believe the referee is referring to immunogenicity of CAR T, which was never a central mechanistic theme of our work. We removed the time-dependent switch in the intrinsic growth rate of CAR T due to immunogenicity in favor of a state-dependent growth rate function that depends on the total lymphocytes ($T = N + C$), which is congruous with our explanation of CAR T dynamics in association with a return to homeostatic normal T cell levels.

for point 2/ by saying they disagree that a deeper analytical / mathematical analysis of the model would be beneficial.

To us, this comment causes some confusion. Both the 3- and 4-compartment model have been analyzed in terms of the fixed points and linear stability analysis. We imagine one might believe that deeper analytic analysis could be achieved by constructing a Lyapunov function to strengthen our claim that the only globally stable fixed point is uncontrolled tumor growth. However, we forgo this approach given a lack of utility of this difficult procedure given the clear long-term dynamics system: as long as CAR T cell numbers decrease, mathematically the only long-term state is tumor (re)growth. Additionally, it seems unnecessary to rule out closed orbits via Poincare-Bendixson, Dulac's criterion or some other method as nothing in this system seems to indicate oscillatory behavior.

An additional feature of the model is the stochastic nature at small tumor sizes, which is only accessible by simulation without further making very strong simplifying assumptions. An analytical discussion of the hybrid or purely stochastic model will be rather technical and will add little to the points made in this manuscript. We believe that this analysis could be beneficial even though it would be based on additional simplifying assumptions, and should be reserved for a future manuscript. Stochastic birth and death processes that involve more than one cell type often pose very hard problems already using simple constant forms for the birth and death rates, and are nearly impossible to analyze using state-dependent rates, which in turn are necessary to model and understand CAR T cell and tumor cell dynamics.

In this manuscript we sought to balance the need for simplifying assumptions and insights into the important transient dynamics, using numerical methods.

3/ by removing the figure related to predicting a new therapy. Also, they discarded the question to assess patient heterogeneity from the data.

Please see our response above regarding re-treatment. In regard to the question of heterogeneity: we did not discount the question but simply noted that sufficient data is unavailable: currently, we do not have a data set available that ties the cell compartments together in individual patients. This is an active area of interest for us and others, which likely will require ongoing funding support and depend upon publication of our current findings and further development of the field.

The main statement remaining in this paper from the original version, is 1/ to provide a simple mathematical model to explain the curves of CAR T cell dynamics versus non-CAR T cells from real patient data when there is B cell depletion before CAR T cell therapy, and (newly added) 2/ to describe stochastically when are the expected times of cure versus relapse although this is based on an 'average patient'. Which is already something adding value to previously published papers (that the authors fairly acknowledge), but lacks a bit the 'meat'.

We thank the referee for acknowledging that our model and approach add value to the current literature. We would like to add that quantitative CAR T cell therapy modeling is a new and rapidly evolving field. In this context, we believe that the model we developed will be of value to future clinical and biological investigations, which should make use of individual patient trajectories once they become available to the public. As highlighted in a recent review on this evolving field (Chaudhury et. al. JCP, 2020), exactly these two points mentioned above, T cell interactions and stochastic tumor behavior, are unique and add to the current state-of-the-art. Importantly, in the LBCL disease setting, cure with CAR T cells is possible without

their persistence—we have attempted to briefly highlight this remarkable fact that is also a feature of our modeling approach (page 8). Together with the current changes and improvements we firmly believe that our manuscript presents novel and relevant quantitative insights.

I want to mention that the work to rewrite the manuscript and keep the track changes is very much appreciated, and the readability was already pretty improved.

Thank you!

I therefore think it would be fair to publish this study but with major substantial modifications.

We appreciate the overall positive assessment of the revision and look forward to addressing the major points below!

Major points

1- Assess patient heterogeneity and its effect into the stochastic predictions of cure. I accept that not all data points are available, but there are methods to cope with it, would it be a bootstrap on the variability of each time-point, or the comparison of the few longitudinal available patients.

We thank the referee for bringing up this important point. It is important to note that the parameter estimation based on a median or a distribution from the quartiles will likely not change much or even at all, as any method of weighted mean squares minimization with loss function will again fit the median, which is why we used median CAR T cell densities. We can, however, attempt to use variable patient trajectories to assess the next point:

This point is critical, as the prediction relies on stochastic variability. In particular, would any patient have the same chance of relapse, or different parameters inferred from different patients would predict different outcome? Meaning: is stochasticity winning over individualized parameters? This would be a critical point for therapy monitoring.

We introduced variability in a 'virtual' patient population by accepting the fits to the median (normal T and CAR T data) as the mean of an underlying distribution with a chosen hyper-parameter that describes this distribution's variance, which led to the distributions discussed in the manuscript. In the long-term, individualized parameters will be key to project a patient's chances of long-term response at early time points.

Indeed, different parameters inferred from different patients *will* lead to a different outcome. We believe that some of this is already answered in the sensitivity analysis shown in Fig. 2 C, but we sought to address this further: We ran additional simulations (now shown in Fig. 3 and described in ESM section 5.4) to assess the impact of changes in parameter values on $\text{prob}(\text{cure})$ and $\text{var}(t_{\text{cure}})$. This analysis shows, stochasticity for a particular patient plays a minor role.

We added a paragraph to the results, now on page 9 that elaborates on this point, and would like to thank the referee for bringing up this important question.

2- Concluding that the carrying capacity of CAR T cells is different than the one of endogenous T cells would need hypothesis comparison. This is true that the model suggests it is possible to explain the dynamics with two different carrying capacity, but does not exclude that other hypothesis could equally well explain the data (especially as the data has not so many time-points). Please thoroughly discuss other possible hypotheses (as the ones I already raised in the first round of revisions), and replace the strong statement by "In the model, different carrying capacities are needed" or "different carrying capacities are consistent with the observed dynamics". - or find ways to better support this statement.

We understand the referee's concern. As a reminder, a previously stated concern was that "the authors should decide whether they want to keep the claim that immunogenicity is the reason [for CAR decay]"

(and support it), *or remove this claim.*” And we had decided to remove this claim, as we feel it cannot be supported at this point.

Another point raised had been that “*Activated T cells produce IL-2, which is a positive feedback for other T cells, so they could actually support each-other. The limiting factor could be that it is hard to access the tumor (especially if the collagen network is impacted in the zones of the tumor). CAR T cell decay could be that they are exhausted or do not manage to access the tumor cells.*” We apologize for not addressing it explicitly before. T cell interactions via secreted factors, tumor microenvironment conditions, as well as exhaustion are very important aspects of CAR T cell dynamics. Each mechanism individually would require additional longitudinal measurements to go beyond the speculative in a model. We feel that in part, our model of normal T-CAR T cell interactions via the carrying capacity factors captures some of the mentioned support, and we now discuss one way to model such explicit interaction (see next paragraph).

Our work now focuses on one possible mechanism that can explain CAR T cell peak *and* decay, but there indeed are others. One could assume that (i) CAR T cells are outcompeted but that there is a form of competitive pressure on normal T cells, leading to temporary coexistence. They could also (ii) be actively supported or cleared by the immune system, via an additional term $\gamma_C * N * C$ (see Supplemental Figure 2). In our final model, we opted for hypothesis (i). Hypothesis (ii) has also now been examined, and we have added our reasoning why we did not consider (ii) beyond initial examination (added to the ESM, Section 3). In addition, variation in stem-ness could be an alternative hypothesis for slow CAR T decay, which we feel goes beyond this study but could be addressed in future models, which must be based on data that explicitly resolves CAR T cell population’s hierarchical structure.

We added respective passages to the Results (page 8), and the Discussion (page 14), which now address these complex topics.

Minor points:

1- Please show the identifiability (profile likelihood?), and parameter correlation as figure, not only in the text

We added the calculated parameter correlations to Figure 3.

2- Figure 1D is identical to the original manuscript, please update with the new model

We apologize for this error. We have updated this with the new model.

3- Supplementary Figure 1 in the ESM is not visible,

This must have been a result of the conversion from pdf->pdf in the electronic editorial manager on the journal website, as the pdf has SI Fig 1 to us prior to electronic submission.

4- Please incorporate better the findings of the ESM into the main manuscript. If not, the ESM is fairly unmentioned although it brings valuable insights.

We have added more mentions of the ESM throughout the main text, with references to specific ESM sections.

5- Define Gompertzian,

We have removed this word and just use Gompertz throughout the text, which we hope is clearer.

6- The results start at Figure 2C, while Figure 2A is already a result. Please take time to explain all Figure 2 in the beginning of the results (eventually, move this part out of the methods)

We agree; done.

Appendix C

Dr. Philipp M. Altrock
Assistant Member
Department of Integrated Mathematical Oncology
H. Lee Moffitt Cancer Center and Research Institute
Tampa, Florida 33612
philipp.altrock@moffitt.org

Sasha Dall, PhD
Editor, Proceedings of the Royal Society B

March 1, 2021

Re: Decision on Manuscript ID RSPB-2021-0229

Dear Dr. Dall,

We are delighted to learn that our manuscript entitled "The roles of T cell competition and stochastic extinction events in CAR T cell therapy" has been accepted for publication in Proceedings B. One of the referees has brought up a few minor comments (in *italics*). These can be seen in the "tracked changes" document.

Referee: 2. Comments to the Author(s).

Again, I must commend the authors for the hard work they have put into this revision and the care in responding to a variety of sometimes contradictory comments. I only have a couple of minor points to make about this revision.

Minor points:

Abstract, Line 30: "before a second CAR T injection"... – done.

Page 4, line 5: Something is out of place with the "both can be transformed" clause. – We clarified what we mean in the first paragraph of the Methods section.

Might be worth starting the Discussion with a bit of context for those readers who skip straight there. – done. This change created some overlap with the next paragraphs, which we thus edited for brevity and clarity.

Page 13, line 1. The sentence starting "Our results" is confusing, and maybe isn't even needed. – We agree and have deleted the short passage.

Page 14, line 9-14. This struck me as a little odd, and either out of place (the last sentence) or not needed. – We agree and have deleted the sentence.

In addition, we applied a few minor cosmetic changes to the manuscript. We want to thank all referees and editors for their much appreciated input and effort.

Sincerely, on behalf of all authors,

Dr rer. nat. Philipp Altrock

Frederick Locke, MD